# Beyond Pairwise Correlations: Higher-Order Redundancies in Self-Supervised Representation Learning

## Abstract

Several self-supervised learning (SSL) approaches have shown that redundancy reduction in the feature embedding space is an effective tool for representation learning. However, these methods consider a narrow notion of redundancy, focusing on pairwise correlations between features. To address this limitation, we formalize the notion of embedding space redundancy and introduce redundancy measures that capture more complex, higher-order dependencies. We mathematically analyze the relationships between these metrics, and empirically measure these redundancies in the embedding spaces of common SSL methods. Based on our findings, we propose Self Supervised Learning with Predictability Minimization (SSLPM), a method for reducing redundancy in the embedding space, as a tool to understand the role of redundancies in SSL. SSLPM combines an encoder network with a predictor engaging in a competitive game of reducing and exploiting dependencies respectively. We demonstrate with SSLPM that while linear redundancy reduction results in competitive performance, we see no evidence of non-linear redundancy reduction being beneficial. Moreover, we find, that the best performing SSL methods exhibit low embedding space redundancy, suggesting that even methods without explicit redundancy reduction mechanisms perform redundancy reduction implicitly.

## 1 Introduction

Self-supervised learning has been shown to produce image embeddings of similar quality as its supervised learning counterparts (Chen et al., 2020). Early methods, such as SimCLR, achieved this by using positive and negative sample pairs within the loss function. A positive pair refers to two different (random) perturbations of the same input image, whereas a negative sample pair refers to perturbed versions of two different input images. The SimCLR loss function encourages similar representations for positive pairs and dissimilar ones for negative pairs. In contrast, subsequent work by Zbontar et al. (2021) replaced negative sample pairs in the loss function with covariance reduction in the representation space. Instead of using negative sample pairs, their loss function reduces the redundancy between representation dimensions by reducing pairwise correlations.

While pairwise decorrelating the representation space produces impressive results (Zbontar et al., 2021), it ignores potential redundancies that might remain. More precisely, the loss function does not explicitly reduce higher-order (involving more than two representation features) and non-linear redundancies in the representation space. In light of this observation, several natural questions arise:

- How can redundancy in SSL embedding spaces be quantified?

- How do redundancies in the embedding space affect downstream performance?

- Can performance be further improved by removing additional, more complex redundancies?

---

*These authors contributed equally to this work.

To answer these research questions we: (1) Introduce formal definitions of embedding space redundancies, specifically pairwise, linear, and non-linear redundancies, and derive their theoretical relationships; (2) Propose Self-Supervised Learning with Predictability Minimization (SSLPM) as a flexible tool for testing the effects of reducing more general forms of redundancy; (3) Empirically investigate the relationship between model performance and embedding space redundancy in a wide range of SSL methods including Barlow Twins (Zbontar et al., 2021), BYOL (Grill et al., 2020), NNCLR (Dwibedi et al., 2021), SimCLR (Chen et al., 2020), MocoV3 (He et al., 2020; Chen et al., 2021), VICReg (Bardes et al., 2021), and VIbCReg (Lee & Aune, 2021).

In our experiments, we find that

- Reducing additional redundancies in training does not result in higher downstream performance.

- Models with explicit redundancy reduction, such as Barlow Twins or SSLPM, show a clear link between performance and linear redundancy. However, this link does not hold in general.

- All methods outperforming our SSLPM model exhibit strictly less embedding space redundancy, suggesting that even methods without explicit redundancy reduction perform redundancy reduction implicitly. However, for SSL methods in general, performance is only weakly correlated with embedding space redundancy.

- The projector [1] depth has a significant impact on redundancy reduction: more projector layers result in less linear and nonlinear redundancy in the embedding space.

## 2 Related Work

Early methods for SSL, such as SimCLR (Chen et al., 2020) or InfoNCE (Oord et al., 2018), use a contrastive loss function based on positive and negative input pairs. The loss function incentivizes augmentations from the same input (positive pair) to produce similar embeddings, while pushing embeddings from different inputs (negative pairs) further apart in the embedding space. Follow-up works on SSL, such as BYOL (Grill et al., 2020) and SiamSiam (Chen & He, 2020), have shown that negative samples are not always necessary by using asymmetric designs in their Siamese networks.

A key challenge in SSL is the prevention of model collapse, whereby all embeddings converge to the same point so as to reduce the positive pair loss. Model collapse leads to no useful representations being learned. Instead of using negative pairs explicitly, Barlow Twins (Zbontar et al., 2021), VicReg (Bardes et al., 2021), VibCReg (Lee & Aune, 2021) and W-MSE (Ermolov et al., 2021) address model collapse by decorrelating features in the representation space of the model. By decorrelating the features in the representation space, redundancy is reduced.

The idea of representation space redundancy reduction was first introduced by Schmidhuber et al. (1996) and Schraudolph et al. (1999) through their concept of predictability minimization. In predictability minimization, an encoder transforms inputs to an $n$-dimensional representation space, on which $n$ predictors try to predict, in a leave-one-out fashion, every single representation feature from the others. The encoder tries to minimize redundancy by ensuring predictors cannot achieve a low prediction loss. The idea of two networks engaging in a competitive game was later revisited in the context of Generative Adversarial Networks (Goodfellow et al., 2014; Schmidhuber, 2020).

Nevertheless, predictors are not novel to SSL and have found applications in (Assran et al., 2023) where they aim to reconstruct representations of surrounding image patches from a single patch.

Moreover, recent work by Shwartz Ziv & LeCun (2024) has drawn connections between compression in SSL methods and information theory. Our work, empirical in nature offers new insights in the role of redundancy reduction in SSL.

---

[1]Typically SSL networks use a projector between the embedding space used for downstream tasks and the representation space where the loss is calculated. The projector network consists of a stack of Feed-forward layers. Please refer to Figure 1.

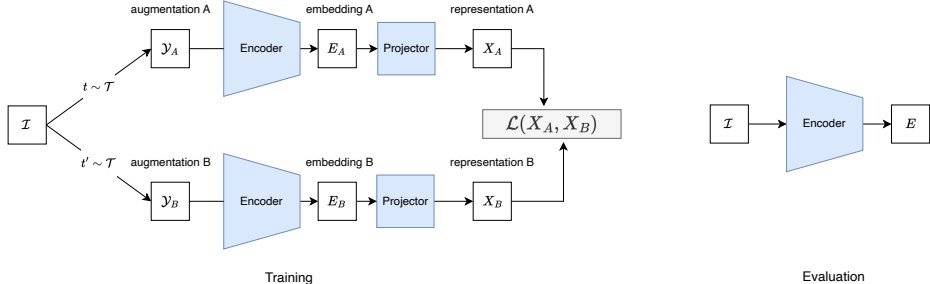

Figure 1: Siamese training set up with encoder in self supervised learning. Embeddings are used in downstream applications, whereas representations are used for loss calculation during pretraining. After pretraining, during evaluation, only the encoder is kept and the randomized augmentations $\tau$ are not applied.

## 3 Methods

### 3.1 Background

Our work is inspired by and builds on Barlow Twins (Zbontar et al., 2021). Barlow Twins is a self-supervised learning approach based on the Siamese training setup depicted in Figure 1. It is used to pretrain neural networks before these are applied to downstream tasks.

In a Siamese network, an input batch $\mathcal{I}$ is passed through two different transformations $\mathbf{t}$ and $\mathbf{t}'$ drawn at random from a distribution of transformations $\mathcal{T}$, producing augmentations $\mathcal{Y}_\mathcal{A}$ and $\mathcal{Y}_\mathcal{B}$. The augmentations are then passed though a neural network, consisting of an encoder and a projector in series, yielding two batches of representations $\mathbf{X_A}$ and $\mathbf{X_B}$.

The Barlow Twins loss function, as displayed in Equation 1, uses these representations to compute the dot product matrix $\mathbf{X_A}^T\mathbf{X_B}$. Before doing so, each dimension of the representations is centered and standardized within its respective batch. The on-diagonal elements of the covariance matrix are then used for the invariance loss $\mathcal{L}_{\text{invariance}}$, whereby each representation dimension is optimized to have a correlation of 1 regardless of the augmentation. Meanwhile, the redundancy reduction term $\mathcal{L}_{\text{redundancy reduction}}$ minimizes the off-diagonal elements of the dot product matrix, which effectively pairwise decorrelates the representation dimensions.

$$\mathcal{L}_{\text{Barlow Twins}} = \underbrace{\sum_i (1 - \left(\mathbf{X_A}^T \cdot \mathbf{X_B}\right)_{ii})^2}_{\mathcal{L}_{\text{invariance}}} + \lambda \underbrace{\sum_i \sum_{j \neq i} \left(\mathbf{X_A}^T \cdot \mathbf{X_B}\right)_{ij}^2}_{\mathcal{L}_{\text{redundancy reduction}}} \tag{1}$$

Similar to contrastive learning (CL) methods, Barlow Twins aims to pull the representations from the same input sample (positive pairs) closer together. In this way, the encoder learns to become invariant to the augmentations in $\mathcal{T}$. This is achieved by forcing the diagonal elements of the correlation matrix towards 1. However, unlike CL methods which push representations from different input samples (negative pairs) further apart, Barlow Twins focuses on reducing the redundancy between the individual features of $\mathbf{X_A}$ and $\mathbf{X_B}$. This is achieved by forcing the off-diagonal of the correlation matrix towards 0.

After the pretraining phase has been completed, the projector is removed and only the encoder is used to embed inputs for downstream tasks. A prediction head, often a single linear layer, is added to produce the desired output format. Fine-tuning is typically done only on the prediction head, and we will also follow this process in this paper.

### 3.2 Redundancy Measures

Inspired by the pairwise correlation redundancy reduction term, we introduce a set of redundancy measures with the aim of capturing additional redundancy between the representation features.

Let $\mathbf{X} = [X_1, \ldots, X_n]^T$ be a vector of $n$ centered random variables, with each variable $X_i$ representing the value of the $i$-th dimension of the $n$ dimensional embedding space. We assume $\mathbb{E}[X_i] = 0$ and $\mathrm{Var}[X_i] = 1$ which is achieved by standardization over the dataset. For convenience we define $\mathbf{X}_{-j} = [X_1, \ldots, X_{j-1}, X_{j+1} \ldots X_n]^T$

**Definition 3.1** (Average Absolute Covariance)**.** For the $n$ dimensional random vector $\mathbf{X}$ with each dimension satisfying $\mathbb{E}[X_i] = 0$ and $\mathrm{Var}[X_i] = 1$, we define the average absolute covariance of dimension $j$ ($\mathrm{AAC}_j$) as well as the total average absolute covariance (AAC) as

$$\mathrm{AAC}_j(\mathbf{X}) := \frac{\sum_{i=1:i\neq j}^{n} |\mathrm{Cov}(X_i, X_j)|}{(n-1)} \qquad \mathrm{AAC}(\mathbf{X}) := \frac{1}{n}\sum_{j=1}^{n} \mathrm{AAC}_j(\mathbf{X}) \qquad (2)$$

Inspired by the concept of predictability minimization by Schmidhuber et al. (1996), we complement the definition of AAC with a definition of predictability, where instead of measuring the pairwise correlation between variables, we measure to what degree a variable can be reconstructed from all others using a *predictor* network.

**Definition 3.2** (Predictability)**.** For the $n$ dimensional random vector $\mathbf{X}$ with each dimension $\mathbb{E}[X_i] = 0$, $\mathrm{Var}[X_i] = 1$, we define the predictability of index $j$ ($\mathrm{Pred}_j$) and the total predictability (Pred) as

$$\mathrm{Pred}_j(X_j|\mathbf{X}_{-j}) := 1 - \min_{F \in \mathcal{F}} \mathbb{E}_{X_i, 1 \leq i \leq n}\left[||F(\mathbf{X}_{-j}) - X_j||_2^2\right] \qquad \mathrm{Pred}(\mathbf{X}) := \frac{1}{n}\sum_{j=1}^{n}\mathrm{Pred}_j(X_j|\mathbf{X}_{-j}) \quad (3)$$

where $\mathcal{F} := \{F : \mathbb{R}^{n-1} \to \mathbb{R}\}$ is the set of all possible functions.

Given this definition, determining predictability is intractable in general since we minimize over the set of all functions $\mathcal{F}$. We therefore introduce two approximations:

- Linear Redundancy (LR) restricts the function space $\mathcal{F}$ to linear functions:

$$\mathrm{LR}_j(\mathbf{X}) = 1 - \min_{\vec{b}} \mathbb{E}\left[||\mathbf{X}_{-j} \cdot \vec{b} - X_j||_2^2\right] \qquad \mathrm{LR}(\mathbf{X}) = \frac{1}{n}\sum_{j=1}^{n}\mathrm{LR}_j(\mathbf{X}) \qquad (4)$$

  with $\vec{b}$ representing the coefficients of a linear regression. In practice, we deviate from this for numerical stability by using ridge regression with penalty $\mu$, with $\mu$ being chosen through cross-validation.

- Non-linear Redundancy (NLR) restricts the function space to neural networks $\mathrm{MLP}_\theta$ parameterized by weights and biases $\theta$.

$$\mathrm{NLR}_j(\mathbf{X}) = 1 - \min_{\theta} \mathbb{E}\left[||\,\mathrm{MLP}_\theta\,(\mathbf{X}_{-j}) - X_j||_2^2\right] \qquad \mathrm{NLR}(\mathbf{X}) = \frac{1}{n}\sum_{j=1}^{n}\mathrm{NLR}_j(\mathbf{X}) \qquad (5)$$

  For our theoretical results, we restrict $\mathrm{MLP}_\theta$ to at least 1 hidden layer and width at least 2 with ReLU activations.

Further details on how the definitions are applied and empirical data is collected are laid out in Appendix A. In our empirical analyses, we effectively use an MLP with two hidden layers of dimensions 128 and 64 respectively.

### 3.3 Mathematical Properties and Relationships

Given the above definitions, we now show that they are all bounded between 0 and 1, and derive how they are theoretically related to one another. The results rely on the assumption that all random variables are centered and have unit variance.

**Lemma 3.1** ($0 \leq \mathrm{AAC}_j(\mathbf{X}) \leq \sqrt{\mathrm{LR}_j(\mathbf{X})} \leq \sqrt{\mathrm{NLR}_j(\mathbf{X})} \leq 1$)**.** For random variables $\mathbf{X} = [X_1, \ldots X_n]^T$ with $\mathrm{Var}[X_i] = 1$ and $\mathbb{E}[X_i] = 0$, for all $i \in [1, n]$ it holds that

- $\text{AAC}_j(\mathbf{X})$, $\text{LR}_j(\mathbf{X})$, and $\text{NLR}_j(\mathbf{X})$ are bounded between 0 and 1.
- $\text{AAC}_j(\mathbf{X}) \leq \sqrt{\text{LR}_j(\mathbf{X})}$
- $\text{LR}_j(\mathbf{X}) \leq \text{NLR}_j(\mathbf{X})$

See proof on page 18.

**Theorem 3.2** $(0 \leq \text{AAC}(\mathbf{X}) \leq \sqrt{\text{LR}(\mathbf{X})} \leq \sqrt{\text{NLR}(\mathbf{X})} \leq 1)$. For random variables $\mathbf{X} = [X_1, \ldots X_n]^T$ with $\text{Var}[X_i] = 1$ and $\mathbb{E}[X_i] = 0$ for all $i \in [1, n]$ it holds that

- $\text{AAC}(\mathbf{X})$, $\text{LR}(\mathbf{X})$, and $\text{NLR}(\mathbf{X})$ are bounded between 0 and 1.
- $\text{AAC}(\mathbf{X}) \leq \sqrt{\text{LR}(\mathbf{X})}$
- $\text{LR}(\mathbf{X}) \leq \text{NLR}(\mathbf{X})$

See proof on page 20.

**Corollary 3.3.** For $X_1, \ldots X_n$ mutually independent variables with all $X_i$ centered unit variance random variables, it follows that $0 = \text{AAC}(\mathbf{X}) = \sqrt{\text{LR}(\mathbf{X})} = \sqrt{\text{NLR}(\mathbf{X})}$

See proof on page 20.

**Corollary 3.4.** For $X_1, \ldots, X_n$ centered random variables with unit variance, it holds that

$$\text{AAC}([X_1, \ldots, X_n]) = 0 \iff \text{LR}([X_1, \ldots, X_n]) = 0 \tag{6}$$

See proof on page 20.

Informally speaking by Corollary 3.4, AAC and LR are similar in terms of the redundancies they can maximally capture, which begs the question of why we distinguish between them; for this let us look at an example:

**Example 3.1.** Let $X_1, \ldots, X_n$ are centered random variables with unit variance with $X_n = \frac{1}{\sqrt{n-1}} X_1 + \ldots + \frac{1}{\sqrt{n-1}} X_{n-1}$ but with $X_1, \ldots X_{n-1}$ mutually independent.

As any variable is completely linearly reconstructable from all other ones, it follows that $\text{LR}([X_1, \ldots, X_n]) = \text{NLR}([X_1, \ldots, X_n]) = 1$.

However, using independence of the first $n-1$ variables and

$$\text{Cov}(X_i, X_n) = \text{Cov}\left(X_i, \frac{1}{\sqrt{n-1}} X_1 + \ldots + \frac{1}{\sqrt{n-1}} X_{n-1}\right) = \frac{\text{Var}[X_i]}{\sqrt{n-1}} = \frac{1}{\sqrt{n-1}} \tag{7}$$

we find

$$\text{AAC}([X_1, \ldots, X_n]) = \frac{0 \cdot \binom{n-1}{2} + \sum_{i=1}^{n-1} \text{Cov}(X_i, X_n)}{\binom{n}{2}} = \frac{\sqrt{n-1}}{\binom{n}{2}} \in \mathcal{O}(n^{-3/2}) \tag{8}$$

meaning that $\text{AAC}([X_1, \ldots, X_n])$ becomes diminishingly small as $n$ increasing.

Hence, whereas AAC measures how strongly variables are *pairwise* correlated, LR and NLR intuitively speaking also capture interactions involving three or more variables. They measure to what degree information about each variable is contained within the others.

Furthermore, the nonlinear nature of NLR allows it to capture more redundancy than LR as Corollary 3.5 demonstrates.

**Corollary 3.5.** For $X_1, \ldots, X_n$ centered variables with unit variance, $\text{LR}([X_1, \ldots, X_n]) = 0 \;\;\not\!\!\!\Longrightarrow\;\; \text{NLR}([X_1, \ldots, X_n]) = 0$

See proof on page 21.

### 3.4 SSLPM

To experimentally probe the role of redundancies in SSL, we introduce Self Supervised Learning with Predictability Minimization (SSLPM) where instead of reducing pairwise correlations as in Barlow Twins (Equation 1) we minimize redundancy via predictability minimization (Schmidhuber et al., 1996). We do this by using a standard Siamese set-up for SSL but extended with a predictor network operating on the concatenated representations from the two augmentations. The loss of the encoder is then given by the invariance loss combined with the new prediction loss coming from the predictor.

The set-up of SSLPM is illustrated in Figure 2.

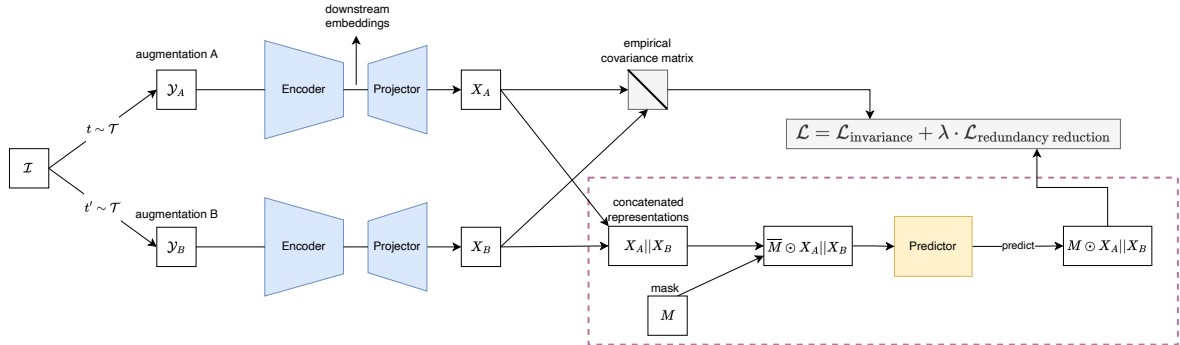

Figure 2: Schematic representation of the SSLPM model with the two actors, the encoder-projector network (blue) and the predictor network (yellow). Our contribution is indicated with the dashed box.

**Predictability Minimization** Since representations can be high-dimensional, we depart from the typical leave-one-out predictability minimization scheme introduced by Schmidhuber et al. (1996) and take inspiration from random masking Devlin et al. (2018); Hsu et al. (2021) originally introduced in language modeling. Instead of performing $n$ leave-one-out predictions, we train a single network to predict randomly masked input features.

For this, let us define $\mathcal{B}$ as a mini batch of $b = |\mathcal{B}|$ images that are fed through the encoder and projector, resulting (after standardization) in two augmented representations, $\mathbf{X_A}$ and $\mathbf{X_B} \in \mathbb{R}^{b \times n}$.

After concatenating them into a single tensor $\mathbf{X_{A||B}} \in \mathbb{R}^{2b \times n}$, we generate a binary mask $\mathbf{M} \in \mathbb{R}^{2b \times n}$ with $\mathbf{M}_{i,j} \sim \mathrm{Bernoulli}(1/2)$, i.e., in expectation half of all input features are masked. In the following, we further denote the inverse mask by $\bar{\mathbf{M}} := 1 - \mathbf{M}$.

Let $R_\theta : \mathbb{R}^n \to \mathbb{R}^n$ be a predictor parametrized by $\theta$. We aim to train $R_\theta$ such that for a randomly sampled mask $\mathbf{M}$, we minimize the average squared prediction error of the masked entries. This can be expressed as the following minimization problem:

$$\min \sum_{i,j} \mathbf{M}_{i,j} \left( R_\theta(\bar{\mathbf{M}} \odot \mathbf{X_{A||B}})_{i,j} - \mathbf{X_{A||B}}_{i,j} \right)^2 \tag{9}$$

$$= \min \left\| \mathbf{M} \odot \left( R_\theta(\bar{\mathbf{M}} \odot \mathbf{X_{A||B}}) - \mathbf{X_{A||B}} \right) \right\|_F^2, \tag{10}$$

where $\odot$ denotes elementwise multiplication. In other words, we aim to predict the masked indices of the concatenated representations from the non-masked indices. Note that we only care about the indices that were masked, reconstruction of the others does not contribute to the loss.

Combining this with the invariance loss used in Barlow Twins (Equation 1) yields the following loss function for SSLPM:

$$\mathcal{L}_{\text{SSLPM}} = \underbrace{\sum_i \left(1 - \left(\mathbf{X_A}^T \cdot \mathbf{X_B}\right)_{ii}\right)^2}_{\mathcal{L}_{\text{invariance}}} - \lambda \cdot \underbrace{\frac{1}{|\mathbf{M}|} \sum_{i,j} \mathbf{M}_{i,j} \left(R_\theta(\mathbf{X_{A||B}})_{i,j} - \mathbf{X_{A||B}}_{i,j}\right)^2}_{\mathcal{L}_{\text{pred}}}, \tag{11}$$

where the prediction loss is adjusted for the number of masked entries $|\mathbf{M}| = \sum_{i,j} \mathbf{M}_{i,j}$, and the relative importance of the two loss terms is controlled by the hyperparameter $\lambda$.

**SSLPM: A Competitive Game**   By construction, the two networks, the encoder and the predictor, engage in a competitive game. The predictor aims to predict the masked features and the encoder aims to (1) ensure corresponding features from different augmentations correlate with each other and (2) make the predictor's task as hard as possible. This is similar to a GAN (Goodfellow et al., 2014) set-up, where the loss of the generator depends on the loss of the discriminator.

**Predictor Design**   In the discussion above we have intentionally not put any constraints on the predictor architecture $R_\theta$. We now introduce the two predictor designs we use in our experiments allowing for a comparison between linear and non-linear redundancies, as well as varying the strength of the latter.

- SSLPM-SGD: Self supervised learning with stochastic gradient descent-based predictability minimization, where the predictor network is optimized using gradient descent, and

- SSLPM-RR: Self supervised learning with ridge regression-based predictability minimization, where a closed form solution for the optimal ridge regression weights is used.

### 3.4.1   SSLPM-SGD

In the gradient descent-based predictor, we use an MLP predictor (once without any hidden layers, and once with two hidden layers) and ReLU activations. After every forward pass of the encoder, the standardized and merged representations $\mathbf{X_{A||B}}$ are used to train the predictor for multiple iterations. Note that the objective of the predictor (Equation 9) is dependent on the mask $\mathbf{M}$. To increase stability, we sample a new random mask for each optimization step. Additionally, we sample a validation mask $\mathbf{M}_V$ at the beginning, which is used for early stopping. Once the optimization of the predictor has concluded, $\mathbf{M}_V$ and the optimized predictor is used to compute the predictability loss for the encoder.

### 3.4.2   SSLPM-RR

If the predictor is a linear layer, we can solve for its optimal weights directly without gradient-based optimization. We define the predictor network as $R_{\boldsymbol{W}}(X) = X\boldsymbol{W}$ where $\boldsymbol{W} \in \mathbb{R}^{k \times k}$ is the weight matrix of a multilinear regression. We deviate slightly from the loss function presented in Equation 9 to one more suited to linear regression and introduce a ridge penalty for the network weights using the following loss function for the predictor:

$$\min \left\| R_{\boldsymbol{W}}(\bar{\mathbf{M}} \odot \mathbf{X_{A||B}}) - \mathbf{X_{A||B}} \right\|_F^2 + \mu \left\| \boldsymbol{W} \right\|_F^2 . \tag{12}$$

This leads to the following closed-form solution for the weights:

$$\boldsymbol{W} = \left( \left( \left(\bar{\mathbf{M}} \odot \mathbf{X_{A||B}}\right)^T \cdot \left(\bar{\mathbf{M}} \odot \mathbf{X_{A||B}}\right) + \mu \boldsymbol{I} \right)^{-1} \cdot \left(\bar{\mathbf{M}} \odot \mathbf{X_{A||B}}\right)^T \cdot \mathbf{X_{A||B}} \right), \tag{13}$$

where $\mu$ is the ridge penalty. The ridge penalty is needed when the dimensionality of the regression is large but the batch size is small. In this case, linear regression would involve inverting a singular matrix, but introducing a Ridge penalty circumvents this problem. More details on the loss function as well as other design choices can be found in Appendices B and C.

### 3.4.3 Implementation Details

Our implementation is based on Tsai et al. (2021) and implemented in the solo-learn (Da Costa et al., 2022) framework. For experimental consistency, all models were (re-)trained on our infrastructure.

For all models, except SSLPM, we use the best performing hyperparameters from the solo-learn repository, which the authors optimized extensively for CIFAR-10 and ImageNet-100.

SSLPM-SGD was optimized on CIFAR-10 with the hyperparameters then also being used on CIFAR-100 and ImageNet-100. SSLPM-RR was optimized on CIFAR-10 and ImageNet-100 in line with the models from solo-learn. Further implementation details can be found in Appendix E.

## 4 Results & Analysis

In the following, we perform a deeper analysis into embedding space redundancies and an investigation into SSLPM. As SSLPM-RR outperforms SSLPM-SGD in our experiments, most analyses are based on the former.

### 4.1 Overall Performance

For all methods used in our work, we report accuracy numbers on the three datasets: CIFAR-10, CIFAR-100, and ImageNet-100. For the CIFAR-10 and CIFAR-100 datasets, we use a ResNet-18 backbone (He et al., 2016) trained for 1000 epochs and report Top 1 and Top 5 accuracies. We use the same methodology as solo-learn (Da Costa et al., 2022) with an on-line linear prediction head that is jointly trained on top of the ResNet-18 backbone during the backbone training. For ImageNet-100 we use a ResNet-18 backbone trained for 400 epochs and report the same Top 1 and Top 5 accuracies. In appendix C.1 additional results for offline evaluation on ImageNet100 are reported.

Barlow Twins, SSLPM-RR, and SSLPM-SGD were trained with three different seeds, and mean scores with standard deviations are reported. For SSLPM-SGD we present SSLPM-SGD 1l results using a single regressive layer MLP predictor (without hidden layers) and SSLPM-SGD 3l results using two hidden layers as well as a final regressive layer MLP predictor. Details on hyperparameters and training can be found in Appendix E.

Table 1: Evaluation of different models on various datasets. **Top1**, **Top2**, and Top3 models are highlighted per dataset.

| Models | CIFAR-10 | | CIFAR-100 | | ImageNet-100 | |
|---|---|---|---|---|---|---|
| | Top 1 | Top 5 | Top 1 | Top 5 | Top 1 | Top 5 |
| Barlow Twins | 92.21 (0.136) | **99.82 (0.026)** | **70.94 (0.075)** | **92.35 (0.042)** | **80.31 (0.099)** | **95.33 (0.147)** |
| BYOL | 93.03 | 99.76 | 71.04 | 92.42 | 76.12 | 93.92 |
| NNCLR | 91.70 | 99.76 | 68.61 | 91.34 | 79.22 | 94.82 |
| SimCLR | 91.23 | 99.76 | 66.63 | 89.58 | 77.38 | 94.02 |
| MocoV3 | 90.40 | 99.77 | 66.94 | 90.21 | 74.72 | 93.02 |
| VICReg | 91.71 | 99.71 | 68.23 | 90.86 | **79.56** | **95.10** |
| VIbCReg | 90.51 | 99.80 | 66.22 | 89.50 | 78.14 | 94.02 |
| **SSLPM-RR (ours)** | **92.70 (0.144)** | **99.85 (0.029)** | 70.46 (0.230) | 92.14 (0.196) | 79.17 (0.310) | 95.02 (0.178) |
| **SSLPM-SGD 1l** | 90.39 (0.136) | 99.72 (0.015) | 67.74 (0.083) | 90.70 (0.240) | 76.19 (0.196) | 93.90 (0.193) |
| **SSLPM-SGD 3l** | 86.69 (0.295) | 99.62 (0.031) | 47.65 (0.201) | 76.59 (0.170) | 64.27 (0.594) | 88.55 (0.600) |

Table 1 shows most previous methods performing similarly well, with MocoV3 falling slightly behind. We also see that SSLPM-RR is competitive with Barlow Twins and other SSL methods. However, SSLPM-SGD with one MLP layer, although being highly similar to SSLPM-RR, falls a few percentage points behind. When the number of layers in SSLPM-SGD is increased from 1 to 3, the gap widens.

To quantify the impact of redundancy reduction, we trained a Barlow Twins model without redundancy reduction ($\lambda = 0$). This essentially yields a SSLPM base model which has accuracies significantly lower than any proposed models with redundancy reduction, as can be seen in Table 2.

Table 2: Ablation showing the importance of the redundancy reduction term.

| Models | CIFAR-10 | | CIFAR-100 | | ImageNet-100 | |
| --- | --- | --- | --- | --- | --- | --- |
| | Top 1 | Top 5 | Top 1 | Top 5 | Top 1 | Top 5 |
| SSLPM-RR | 92.70 (0.144) | 99.85 (0.029) | 70.46 (0.230) | 92.14 (0.196) | 79.17 (0.310) | 95.02 (0.178) |
| SSLPM-SGD 1l (ours) | 90.39 (0.136) | 99.72 (0.015) | 67.74 (0.083) | 90.70 (0.240) | 76.19 (0.196) | 93.90 (0.193) |
| SSLPM-SGD 3l (ours) | 86.69 (0.295) | 99.62 (0.031) | 47.65 (0.201) | 76.59 (0.170) | 64.27 (0.594) | 88.55 (0.600) |
| SSLPM ($\lambda = 0$) | 77.71 (0.106) | 98.95 (0.060) | 34.05 (1.081) | 63.74 (1.358) | 50.81 (0.388) | 79.86 (0.046) |

These results show that:

1. More general forms of redundancy reduction can work just as well as the covariance-based redundancy reduction used in Barlow Twins.

2. Reducing higher-order redundancies (e.g. SSLPM-SGD-3l) can result in lower performance, though still outperforming the $\lambda = 0$ baseline.

## 4.2 The Link between Performance and Linear Redundancy Reduction

The hyperparameter $\lambda$ as introduced in Equation 11 plays a crucial role in weighing the different losses against one another. Figure 3 shows final Top-1 accuracy in the left plot and the predictability loss in the final epoch of training in the right plot. Although we see increasing $\mathcal{L}_{pred}$ for increasing $\lambda$, optimal performance is reached before maximal predictability loss is reached and falls sharply for $\lambda > 0.25$. More detailed plots on how this evolves during training can be found in Appendix C.2.

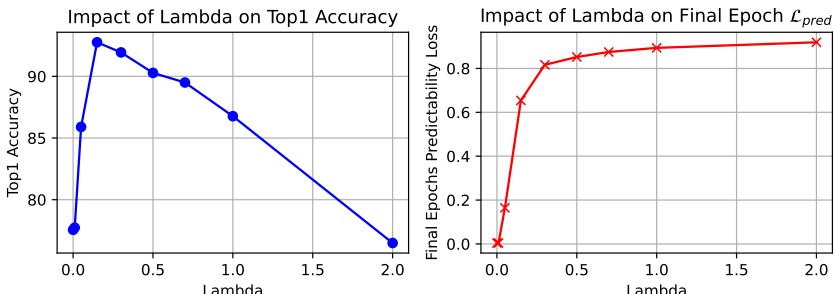

Figure 3: Impact of $\lambda$ in Equation 11 on CIFAR-10.

The observations from Figure 3 indicate that too much redundancy reduction reduces embedding quality. However, stronger emphasis on redundancy reduction in the loss must not result in less redundancy in the final embeddings.

For this, given that different $\lambda$'s result in different performance, we set out to look at these results from a different angle: How does final model performance relate to the redundancy in the embedding space? In Figure 4 we see that for LR there is a clear link, whereby less redundancy results in higher performance (with $p < 0.01$ across all data sets and models). However, for AAC and NLR there seems to be an optimal level of redundancy after which performance degrades. Similar results can be seen in Figure 13 in the appendix for the CIFAR-100 and ImageNet-100 datasets.

From these results one can infer that reducing the redundancy in Barlow Twins and SSLPM-RR aids performance. However, this is not necessarily achieved by increasing the hyperparameter $\lambda$. At a certain point the redundancy reduction term is too strong compared to the invariance loss leading to a drop in performance.

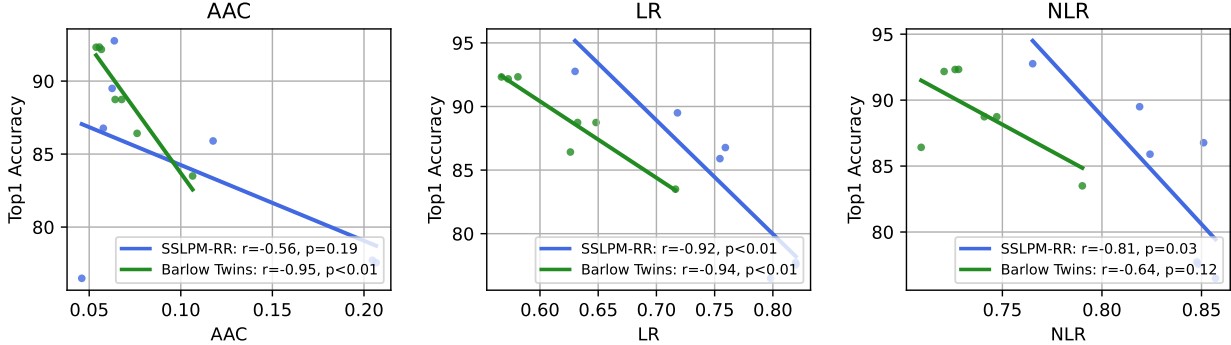

Figure 4: Relationship between Top-1 accuracy and three different redundancy measures on CIFAR-10 plotted with the lines of best fit. For the best fit lines, the Pearson correlation and the $p$-value for the null hypothesis that the data is uncorrelated is reported. LR is the only redundancy measure where the correlation is significant (with $p < 0.01$) for both models.

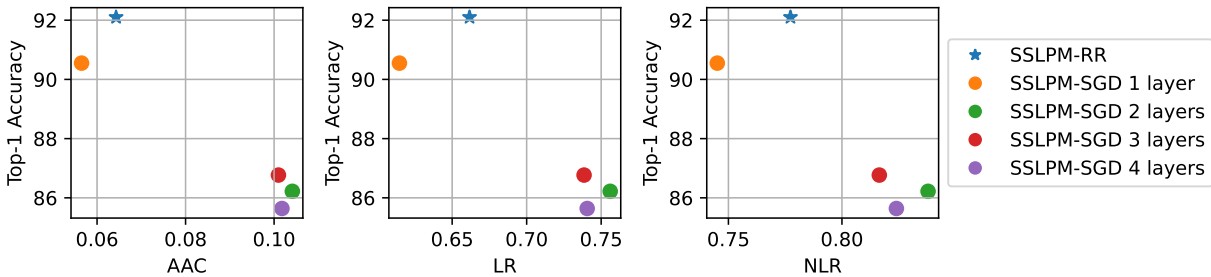

Figure 5: Ablation on the number of layers of the predictor in SSLPM-SGD compared with SSLPM-RR. All results are calculated on CIFAR-10.

### 4.3 Minimizing Non-Linear Redundancy

In comparison to SSLPM-RR, where we have a fixed regression problem to solve in the predictor, we can freely design the predictor in SSLPM-SGD. In Figure 5 we see that (1) all SSLPM-SGD versions underperform SSLPM-RR in terms of accuracy, but the 1 layer SSLPM-SGD version features the least redundancy for all measures, and (2) when the predictor in SSLPM-SGD contains two or more layers, meaning that the predictor can capture non-linearities, there is a steep drop in accuracy as well as the redundancy that is removed. We therefore conclude that there is no evidence that reducing higher-order redundancies is beneficial for downstream embedding performance - in fact we see evidence to the contrary, with performance and actual redundancy being removed falling as the number of layers of the predictor increases.

### 4.4 Projector Depth Impacts Redundancy

The projector, which is an MLP that sits between the embedding space and the space on which the loss is applied, as seen in Figure 2, has been shown to be of crucial importance for many SSL methods (Gupta et al., 2022) where no projector or a less expressive projector usually results in lower accuracy.

In Figure 6 we have measured the three redundancy values for Barlow Twins and SSLPM-RR on CIFAR-10 for different projector depths. Interestingly, we find that SSLPM-RR accuracy, compared to Barlow Twins, is less affected by the number of layers in the projectors. Further, we find it noteworthy that although AAC reveals no clear trends regarding redundancy and accuracy, we see a very clear trend for LR and NLR. As the number of projector layers is increased, the amount of redundancy removed from the embeddings increases. This extends work by (Gupta et al., 2022) and provides a novel perspective on the effects of projectors.

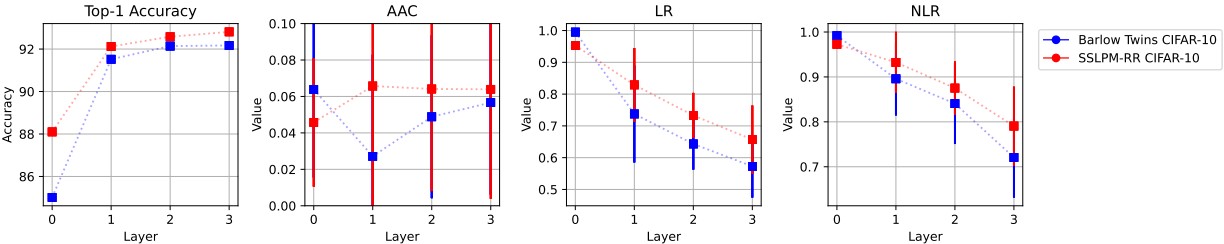

Figure 6: Ablation on the number of layers of the projector for CIFAR-10. Redundancy measures have the standard deviation over all measured indices shown.

Figure 15 in Appendix C show analogous results on CIFAR-100 and ImageNet-100.

## 4.5 Redundancy Across SSL Methods

Given the strong relationship between redundancy and performance for Barlow Twins and SSLPM, one might be tempted to think that low redundancy is a feature of *all* high quality embeddings, whatever the pretraining approach. The evidence presented in Figure 7 for CIFAR-10 and Figure 14 in appendix F for CIFAR-100 does not support this conclusion. P values of 0.79 and 0.93 indicate that likely no linear relationship is present (looking at LR vs Top-1 accuracy). Further, the results are inconsistent across the redundancy measures.

However, in the case of ImageNet-100, also in Figure 14, the data tells a different story with negative correlations between the Top-1 accuracy and all redundancy measures and low p values ($p < 0.05$ for LR and NLR).

From the provided evidence, we cannot conclusively say that embeddings from other training approaches will also exhibit a negative correlation between redundancy and performance.

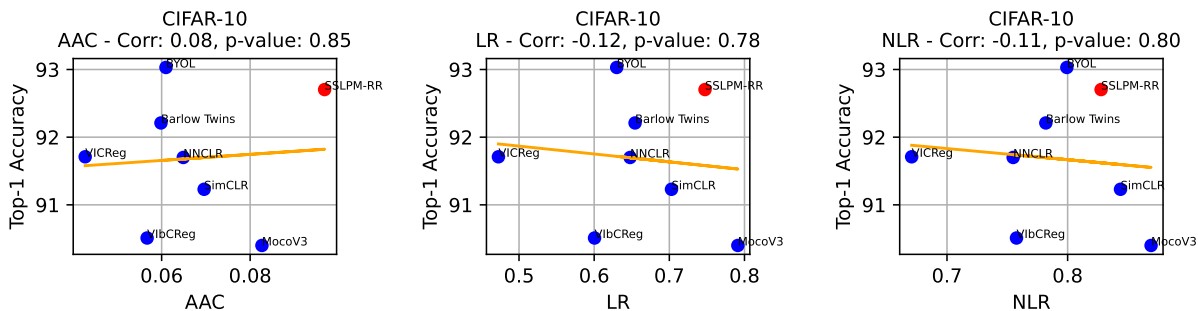

Figure 7: AAC, LR, and NLR embedding space redundancies plotted against Top-1 accuracy for different SSL methods on CIFAR-10. For the best fit lines, Pearson correlation and $p$-value for the null hypothesis that the data is uncorrelated, is reported.

Nevertheless, across all datasets, we consistently find that all methods outperforming SSLPM-RR, namely BYOL, Barlow Twins, VICReg, and NNCLR, exhibit significantly less redundancy in their embeddings. Given that these methods do not perform active redundancy reduction, we see this as evidence that high performing models perform implicit redundancy reduction.

## 5 Conclusion

Although the idea of embedding space redundancies goes back to Schmidhuber et al. (1996), contemporary work on SSL only considers a narrow notion of redundancy. With our contribution, we aim to close this gap

in the SSL literature by introducing a framework for quantifying and studying redundancy in embedding spaces.

We introduce a hierarchy of redundancy measures and establish their theoretical foundations. Furthermore, we conduct an extensive empirical study to investigate the relationship between downstream model performance and embedding space redundancy. We find evidence for an inverse relationship between redundancy and performance when removing linear redundancy. With the introduction of SSLPM, we demonstrate the feasibility of reducing higher-order redundancy relations. However, we find that there is currently no evidence that reducing more complex redundancies is beneficial for downstream model performance questioning the intuitive idea that stronger redundancy reduction is desirable.

## 6 Limitations and Future Work

Despite pairwise correlations (AAC) only offering a narrow view of redundancy in embedding spaces, we have not found evidence that reducing more general redundancies, such as LR (SSLPM-RR) or NLR (SSLPM-SGD), results in better performance. Our empirical results suggest that there is a limit to the redundancy that can be extracted as when using a large weight (large $\lambda$) for the redundancy term in the loss function, both Barlow Twins and SSLPM suffer from model collapse leading to degrading performance.

Future work could explore whether similar results can be found for other data modalities, such as audio, text or multi-modal SSL.

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

## A  Details on Measuring Redundancy

For all trained models, we have extracted the embeddings of all test and train samples. The two datasets were then individually standardized such that every embedding feature has a mean of 0 and a standard deviation of 1.

Hence, let $\vec{X} \in \mathbb{R}^k$ be the output of an encoder, for example the $k = 512$ dimensional output of a ResNet-18, right before the last fully-connected layer. For practical purposes, we always standardize a batch of $n$ embeddings $\mathbf{X} \in \mathbb{R}^{n \times k}$ such that for all dimensions $i$ we have $\mathbb{E}[(X)_i] = 0$ and $\mathrm{Var}[(X)_i] = 1$. Given the standardized embeddings, we define the dot product matrix $\mathbf{C} := \frac{\mathbf{X}^T \cdot \mathbf{X}}{n}$.

For AAC, the values from the dot product matrix of the standardized embeddings are used. For LR and NLR a random sample of 20% of all neurons/features is taken (20% of $512 \approx 102$). Then using 80% of the data, a model was trained to predict the neuron's activation and then evaluated on the remaining 20%.

For NLR, we used a three layer MLP with a fixed number of training steps and for LR ridge regression where we optimized over the ridge penalty.

In the figures, we report the mean of the measures over all neurons predicted.

The redundancy measurement experiments used a NVIDIA RTX 3090 card and took between 30 minutes and 2 hours per dataset and method.

## B  Modification of SSLPM-RR Objective

In our main analysis for SSLPM-RR we have defined $\boldsymbol{W}$ as

$$\boldsymbol{W}_{\mathrm{default}} = \arg\min_{\boldsymbol{W}} \left\| R_{\boldsymbol{W}}(\bar{\mathbf{M}} \odot \mathbf{X}_{\mathbf{A}||\mathbf{B}}) - \mathbf{X}_{\mathbf{A}||\mathbf{B}} \right\|_F^2 + \mu \left\| \boldsymbol{W} \right\|_F^2 \tag{14}$$

with

$$\boldsymbol{W}_{\mathrm{default}} = \left( \left( (\bar{\mathbf{M}} \odot \mathbf{X}_{\mathbf{A}||\mathbf{B}})^T \cdot (\bar{\mathbf{M}} \odot \mathbf{X}_{\mathbf{A}||\mathbf{B}}) + \mu \boldsymbol{I} \right)^{-1} \cdot (\bar{\mathbf{M}} \odot \mathbf{X}_{\mathbf{A}||\mathbf{B}})^T \cdot \mathbf{X}_{\mathbf{A}||\mathbf{B}} \right) \tag{15}$$

which enforces $\boldsymbol{W}$ to reconstruct the complete representations $\mathbf{X}_{\mathbf{A}||\mathbf{B}}$ from the masked representations. In particular, this means, that non-masked entries should be kept as is (i.e. ideally an identity function applied to them).

We can change this and require that we only reconstruct the masked representations and map all non-masked representations to zero, that way we no longer incentivize learning an identity mapping.

This results in the following optimization problem for $\boldsymbol{W}$:

$$\boldsymbol{W}_0 = \arg\min_{\boldsymbol{W}} \left\| R_{\boldsymbol{W}}(\bar{\mathbf{M}} \odot \mathbf{X}_{\mathbf{A}||\mathbf{B}}) - \mathbf{M} \odot \mathbf{X}_{\mathbf{A}||\mathbf{B}} \right\|_F^2 + \mu \left\| \boldsymbol{W} \right\|_F^2 \tag{16}$$

with

$$\boldsymbol{W}_0 = \left( \left( (\bar{\mathbf{M}} \odot \mathbf{X}_{\mathbf{A}||\mathbf{B}})^T \cdot (\bar{\mathbf{M}} \odot \mathbf{X}_{\mathbf{A}||\mathbf{B}}) + \mu \boldsymbol{I} \right)^{-1} \cdot (\bar{\mathbf{M}} \odot \mathbf{X}_{\mathbf{A}||\mathbf{B}})^T \cdot (\mathbf{M} \odot \mathbf{X}_{\mathbf{A}||\mathbf{B}}) \right) \tag{17}$$

During training on CIFAR-10 we barely see a difference between the two models as shown in Table 3. They feature a nearly identical final predictability loss as well as accuracies on CIFAR-10. Hence, we conclude that the exact choice of the objective for the ridge regression is likely not of crucial importance for our method.

## C  Further Results & Analysis

### C.1  Online vs Offline Evaluation

Following the methodology by Da Costa et al. (2022) we report online as well as offline results for ImageNet-100. While in the online case a linear prediction head is also trained during pre-training, bur without gradient

Table 3: Change of regression objective

| Models | CIFAR-10 | | |
| --- | --- | --- | --- |
| | Top 1 | Top 5 | Last Epoch Avg $\mathcal{L}_{pred}$ |
| SSLPM-RR with $\boldsymbol{W}_{\text{default}}$ | 92.81 | 99.87 | 0.65 |
| SSLPM-RR with $\boldsymbol{W}_0$ | 92.75 | 99.82 | 0.66 |

flowing through the prediction head into the encoder, the off-line linear prediction head is trained for 100 epochs on the training data after the backbone has finished (pre-)training and its weights have been completely frozen. Unexpectedly, the differences between the two evaluation modes is within the margin of error.

Table 4: Evaluation of different models on various datasets. **Top1**, **Top2**, and Top3 models are highlighted per dataset.

| Models | ImageNet-100 (Online) | | ImageNet-100 (Offline) | |
| --- | --- | --- | --- | --- |
| | Top 1 | Top 5 | Top 1 | Top 5 |
| Barlow Twins | **80.31 (0.099)** | **95.33 (0.147)** | **80.24 (0.179)** | **95.21 (0.058)** |
| BYOL | 76.12 | 93.92 | 76.66 | 93.74 |
| NNCLR | 79.22 | 94.82 | 79.58 | 94.9 |
| SimCLR | 77.38 | 94.02 | 77.50 | 93.86 |
| MocoV3 | 74.72 | 93.02 | 74.46 | 92.86 |
| VICReg | **79.56** | **95.10** | 79.28 | 94.52 |
| VIbCReg | 78.14 | 94.02 | 78.26 | 93.96 |
| **SSLPM-RR (ours)** | 79.17 (0.310) | 95.02 (0.178) | **79.99 (0.711)** | **94.95 (0.253)** |
| **SSLPM-SGD 1l** | 76.19 (0.196) | 93.90 (0.193) | 76.73 (0.320) | 93.83 (0.147) |
| **SSLPM-SGD 3l** | 64.27 (0.594) | 88.55 (0.600) | 67.93 (0.938) | 90.79 (0.659) |

Table 5: Ablation showing the importance of the redundancy reduction term.

| Models | ImageNet-100 (Online) | | ImageNet-100 (Offline) | |
| --- | --- | --- | --- | --- |
| | Top 1 | Top 5 | Top 1 | Top 5 |
| SSLPM-RR | 79.17 (0.310) | 95.02 (0.178) | 79.99 (0.711) | 94.95 (0.253) |
| SSLPM-SGD 1l (ours) | 76.19 (0.196) | 93.90 (0.193) | 76.73 (0.320) | 93.83 (0.147) |
| SSLPM-SGD 3l (ours) | 64.27 (0.594) | 88.55 (0.600) | 67.93 (0.938) | 90.79 (0.659) |
| SSLPM ($\lambda = 0$) | 50.81 (0.388) | 79.86 (0.046) | 57.78 (0.597) | 84.23 (0.423) |

## C.2 Training Behavior with Different Loss Weights in SSLPM

The hyperparameter $\lambda$ as introduced in Equation 11 plays a crucial role in weighing the different losses against one another. In Figure 8 we plot different loss metrics over training for different $\lambda$'s. We see that a $\lambda$ that is too small (such as 0.01) performs similar to the case where we have $\lambda = 0$. For $\lambda = 1, 2$ we see that $\mathcal{L}_{\text{pred}}$ is close to 1, but this does not lead to the best observed performance. Interestingly, we find that over the course of training even though $\mathcal{L}_{\text{SSLPM}}$ decreases steadily, first $\mathcal{L}_{\text{invariance}}$ nearly vanishes, but then increases again as $\mathcal{L}_{\text{pred}}$ is optimized. This dynamic is especially present for large $\lambda$.

The observations from Figure 8 allow us to conclude that there seems to be a tradeoff between $\mathcal{L}_{\text{invariance}}$ and $\mathcal{L}_{\text{pred}}$ whereby too much redundancy reduction hurts model performance.

## C.3 Effect of Ridge Penalty and Batch Size in SSLPM-RR

When the batch size smaller than half the size of the representation dimension on which we calculate the loss, standard linear regression can no longer be performed as the system is singular. To circumvent this and to increase regression stability, we introduce a ridge penalty. The experiment visualized in Figure 9 uses a

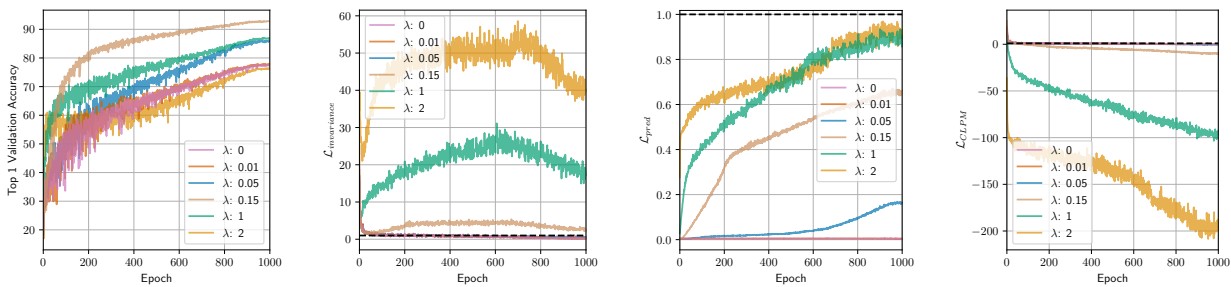

Figure 8: Training dynamics for different $\lambda$: (1) accuracy. (2) $\mathcal{L}_{\text{invariance}}$ (3) $\mathcal{L}_{\text{pred}}$. (4) $\mathcal{L}_{\text{SSLPM}}$.

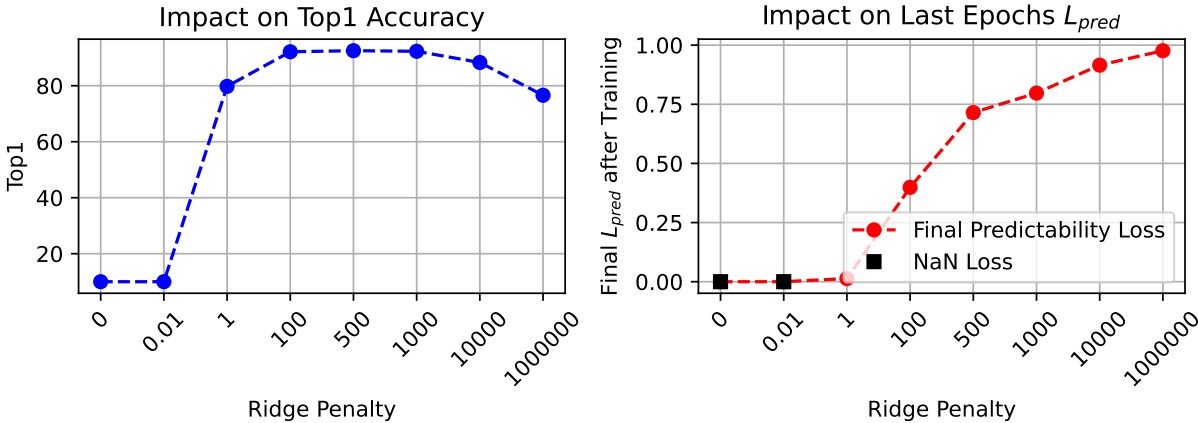

Figure 9: Impact of ridge penalty on Top-1 CIFAR-10 accuracy.

constant batch size of 64 and increasing ridge penalties. We see that a nonexistent or too small ridge penalty results in model collapse due to the regression being singular. However, an appropriately chosen ridge penalty allows the model to learn as if it was operating in a non-singular setting.

Additionally, we show in Table 6 that SSLPM-RR performs well regardless of the batch size as long as an appropriate ridge penalty is chosen.

Table 6: Top-1 accuracy for different batch sizes on CIFAR-10.

| Batch Size | Top 1 Accuracy |
| --- | --- |
| 32 | 91.67 |
| 64 | 92.51 |
| 128 | 92.96 |
| 256 | 92.76 |

## C.4  Impact of Masking Fraction on SSLPM-RR

The masking fraction, the percentage of representation's features to mask during training, is a key hyperparameter in SSLPM. In Figure 10 we have trained multiple different models on CIFAR-10 while varying the masking fraction. We see that as long as the masking fraction is neither too small nor too large, training outcomes are not negatively impacted. Interestingly, masking 5% performed significantly worse than masking 95% of the features.

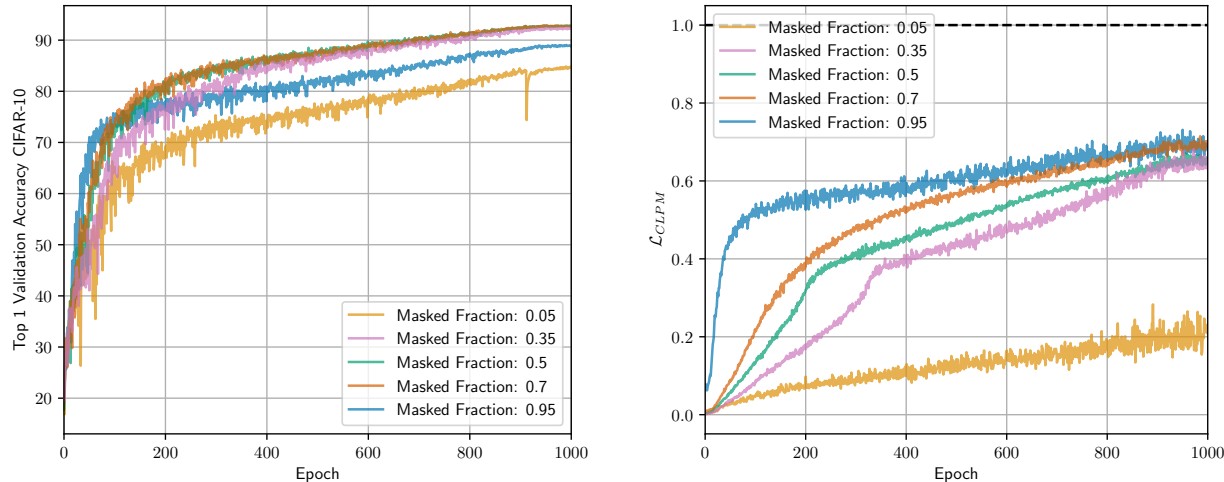

Figure 10: (left) Accuracy over Training. (right) $\mathcal{L}_{\text{pred}}$ over Training

## C.5 The Empirical Relationship Between the Redundancy Measures

Knowing how the redundancy measures are related in theory leaves the question of how the redundancy measures correlate in practice. To answer this question, we plot the redundancies, AAC, LR, and, NLR in pairwise fashion in Figure 11 for all three datasets under investigation, CIFAR-10, CIFAR-100, and ImageNet-100.

From Figure 11 we learn that the redundancy definitions correlate well, especially the more related ones. AAC and LR have a correlation coefficient of 0.90 whereas LR and NLR have a correlation coefficient of 0.94.

Not surprisingly we find that for a given level of AAC redundancy we find higher levels of LR as well as for a given level of LR we find a higher level of NLR. This empirically validates our intuition with LR's ability to be high despite low AAC as well as NLR capturing nonlinear redundancies that LR cannot.

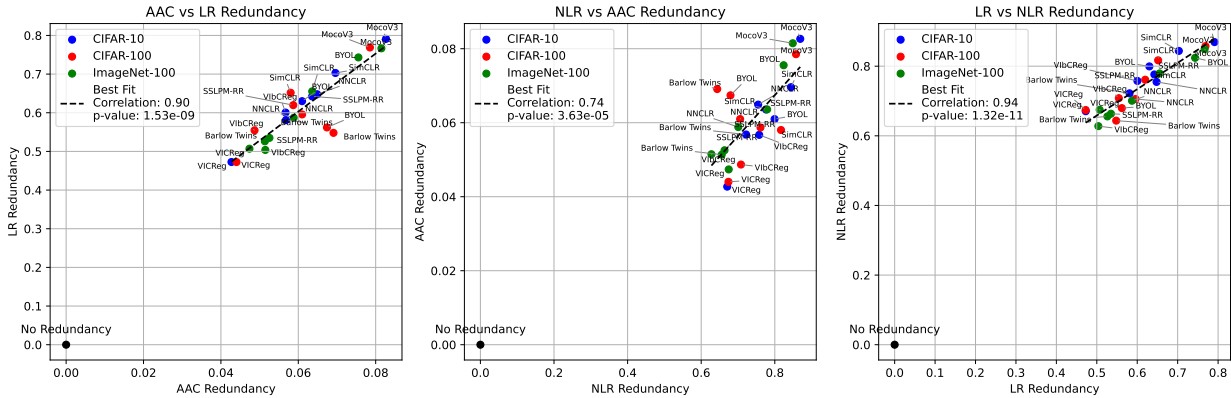

Figure 11: AAC, LR, and NLR plotted against each other for test set embeddings of different models on CIFAR-10, CIFAR-100, and ImageNet-100 datasets.

## C.6 Redundancies in the Test and Train Sets

So far, we have only looked at embeddings from test datasets. However, it begs the question if the results we see on the test sets also hold for the training dataset. To investigate this, we have collected the average

redundancy values for all models under consideration and plotted them onto a pairwise redundancy chart. In Figure 12 we see the CIFAR-10 train and test embeddings colored differently together with two best fit lines. We see that for a given level of AAC redundancy, there is more LR redundancy in the test than the train set. The same holds for NLR, where for a given level of LR redundancy we find more NLR redundancy in the test set. Figure 16 shows that these findings transfer over to CIFAR-100 and ImageNet-100.

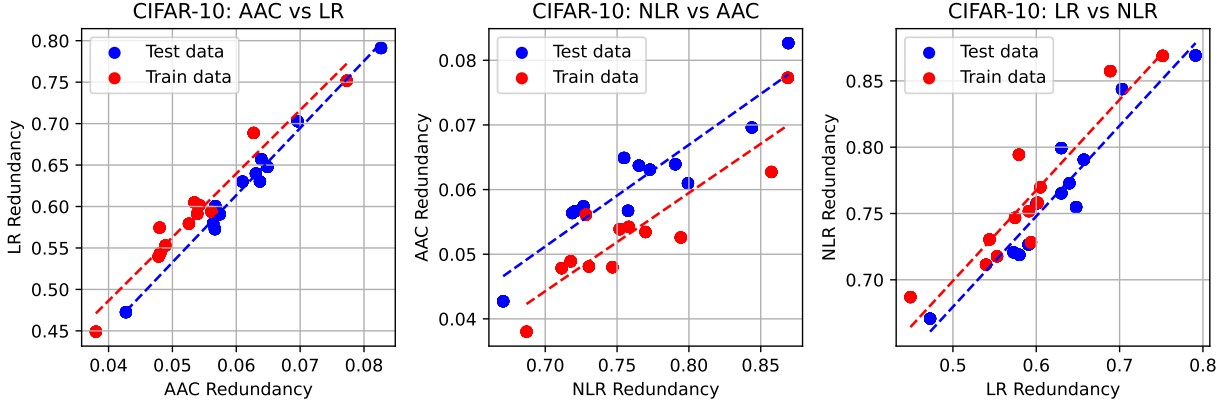

Figure 12: Difference in redundancy between test and train sets for CIFAR-10 embeddings.

## D    Omitted Proofs

**Lemma 3.1** $(0 \leq \mathrm{AAC}_j(\mathbf{X}) \leq \sqrt{\mathrm{LR}_j(\mathbf{X})} \leq \sqrt{\mathrm{NLR}_j(\mathbf{X})} \leq 1)$. For random variables $\mathbf{X} = [\mathrm{X}_1, \dots \mathrm{X}_n]^T$ with $\mathrm{Var}[\mathrm{X}_i] = 1$ and $\mathbb{E}[\mathrm{X}_i] = 0$, for all $i \in [1, n]$ it holds that

- $\mathrm{AAC}_j(\mathbf{X})$, $\mathrm{LR}_j(\mathbf{X})$, and $\mathrm{NLR}_j(\mathbf{X})$ are bounded between 0 and 1.

- $\mathrm{AAC}_j(\mathbf{X}) \leq \sqrt{\mathrm{LR}_j(\mathbf{X})}$

- $\mathrm{LR}_j(\mathbf{X}) \leq \mathrm{NLR}_j(\mathbf{X})$

*Proof of Lemma 3.1.* We show the individual inequalities separately:

$(0 \leq \mathrm{AAC}_j(\mathbf{X}))$: By definition of $\mathrm{AAC}_j(\mathbf{X})$ we have

$$\mathrm{AAC}_j(\mathbf{X}) = \frac{\sum_{i=1:i \neq j}^{n} |\mathrm{Cov}(\mathrm{X}_i, \mathrm{X}_j)|}{(n-1)} \geq 0 \tag{18}$$

as any sum over absolute values is greater or equals 0.

$(\mathrm{AAC}_j(\mathbf{X}) \leq \sqrt{\mathrm{LR}_j(\mathbf{X})})$: The equivalent $\mathrm{AAC}_j(\mathbf{X})^2 \leq \mathrm{LR}_j(\mathbf{X})$ follows from

$$\mathrm{LR}_j(\mathbf{X}) = 1 - \min_{\vec{b}} \mathbb{E}\left[||\mathbf{X}_{-j} \cdot \vec{b} - \mathbf{X}_j||_2^2\right] = 1 - \min_{\vec{b}} \mathbb{E}\left[\left(\sum_{i=1, i \neq j}^{n} (\mathrm{X}_i b_i) - \mathrm{X}_j\right)^2\right] \tag{19}$$

$$= 1 - \min_{\vec{b}} \left(\sum_{i=1, i \neq j}^{n} \sum_{l=1, l \neq j}^{n} \mathrm{Cov}(\mathrm{X}_i, \mathrm{X}_l) b_i b_l - 2 \sum_{i=1, i \neq j}^{n} \mathrm{Cov}(\mathrm{X}_i, \mathrm{X}_j) b_i + \mathrm{Var}[\mathrm{X}_j]\right) \tag{20}$$

Let $k := \mathrm{argmax}_{1 \le k \le n, k \ne j} |\mathrm{Cov}(X_k, X_j)|$. we now arbitrarily choose $b_l = \begin{cases} |\mathrm{Cov}(X_k, X_j)| & l = k \\ 0 & \text{else} \end{cases}$ which allows

$$1 - \min_{\vec{b}} \left( \sum_{i=1, i \ne j}^{n} \sum_{l=1, l \ne j}^{n} \mathrm{Cov}(X_i, X_l) b_i b_l - 2 \sum_{i=1, i \ne j}^{n} \mathrm{Cov}(X_i, X_j) b_i + \mathrm{Var}[X_j] \right) \tag{21}$$

$$\ge - \underbrace{\mathrm{Cov}(X_k, X_k)}_{\mathrm{Var}[X_k] = 1} |\mathrm{Cov}(X_k, X_j)|^2 + 2\mathrm{Cov}(X_k, X_j)|\mathrm{Cov}(X_k, X_j)| = \mathrm{Cov}(X_k, X_j)^2 \tag{22}$$

Meanwhile we also know that

$$\mathrm{AAC}_j(\mathbf{X})^2 = \left( \frac{\sum_{i=1: i \ne j}^{n} |\mathrm{Cov}(X_i, X_j)|}{(n-1)} \right)^2 \le \left( \max_{1 \le i \le n, i \ne j} \mathrm{Cov}(X_i, X_j) \right)^2 \tag{23}$$

combining the two inequalities derived above yields $(\mathrm{AAC}_j(\mathbf{X}) \le \sqrt{\mathrm{LR}_j(\mathbf{X})})$ as desired.

$\underline{(\mathrm{LR}_j(\mathbf{X}) \le \mathrm{NLR}_j(\mathbf{X}))}$:

We show that any linear map $f : \mathbb{R}^n \to \mathbb{R}$ with $f(\vec{x}) = \vec{x} \cdot \vec{b}$, where $\vec{b}$ is the vector of coefficients, can also be represented by an MLP using ReLU with at least one hidden layer with width at least 2.

For the following discussion, we set all biases to 0. Let $h_{i,j}$ be the value at neuron $j$ in hidden layer $i$ after the application of the activation function.

By setting $W_{h_{1,1}} = \beta$ and $W_{h_{1,2}} = -\beta$, where $W_h$ is the weights' matrix for a neuron $h$ which is used for multiplying with incoming activations yields

- $h_{1,1} = \max\left( \vec{x} \cdot \vec{b}, 0 \right)$

- $h_{1,2} = \max\left( -\vec{x} \cdot \vec{b}, 0 \right)$.

For any following layer except the final regressive layer we set $W_{h_{i \ge 2, 1}} = [1, 0]$ and $W_{h_{i \ge 2, 2}} = [0, 1]$ yielding

- $h_{i,1} = \max\left( \vec{x} \cdot \vec{b}, 0 \right)$

- $h_{i,2} = \max\left( -\vec{x} \cdot \vec{b}, 0 \right)$.

For the final regressive layer we use weights [1,-1] which yields a final output of

$$\max\left( \vec{x} \cdot \vec{b}, 0 \right) - \max\left( -\vec{x} \cdot \vec{b}, 0 \right) = \vec{x} \cdot \vec{b} \tag{24}$$

as desired. This shows that any MLP with at least one hidden layer and width 2 can perfectly reconstruct any linear map

For an MLP with width larger than 2 we choose an arbitrary subset of two neurons per layer to embed our constructed MLP from above and set all other weights to 0.

Hence, any linear mapping can be represented by an MLP with width at least 2 and at least 1 hidden layer. From this and the definition from NLP it follows that $\min_b ||\mathbf{X}_{-j} \cdot b - \mathbf{X}_j||_2^2 \ge \min_\theta ||R_\theta(\mathbf{X}_{-j}) - \mathbf{X}_j||_2^2$ as desired.

$\underline{(\mathrm{NLR}_j(\mathbf{X}) \le 1)}$: Clearly $\min_\theta ||R_\theta(\mathbf{X}_{-j}) - \mathbf{X}_j||_2^2 \ge 0$ as the square delimits the expression from below at 0. From this it directly follows that $\mathrm{NLR}_j(\mathbf{X}) = 1 - \min_\theta \mathbb{E}\left[ ||R_\theta(\mathbf{X}_{-j}) - \mathbf{X}_j||_2^2 \right] \le 1$.

From the proven individual inequalities, the claimed results follow. $\qquad \square$

**Theorem 3.2** $(0 \leq \text{AAC}(\mathbf{X}) \leq \sqrt{\text{LR}(\mathbf{X})} \leq \sqrt{\text{NLR}(\mathbf{X})} \leq 1)$**.** For random variables $\mathbf{X} = [X_1, \dots X_n]^T$ with $\text{Var}[X_i] = 1$ and $\mathbb{E}[X_i] = 0$ for all $i \in [1, n]$ it holds that

- $\text{AAC}(\mathbf{X})$, $\text{LR}(\mathbf{X})$, and $\text{NLR}(\mathbf{X})$ are bounded between 0 and 1.

- $\text{AAC}(\mathbf{X}) \leq \sqrt{\text{LR}(\mathbf{X})}$

- $\text{LR}(\mathbf{X}) \leq \text{NLR}(\mathbf{X})$

*Proof of Theorem 3.2.* This follows directly applying Lemma 3.1 $n$ times (for every summand) and using the fact that we can upper bound an arithmetic mean by a quadratic mean.

$$\text{AAC}(\mathbf{X}) = \frac{1}{n} \sum_{j=1}^{n} \text{AAC}_j(\mathbf{X}) \geq \frac{1}{n} \sum_{j=1}^{n} 0 = 0 \tag{25}$$

$$\text{AAC}(\mathbf{X}) = \frac{1}{n} \sum_{j=1}^{n} \text{AAC}_j(\mathbf{X}) \leq \frac{1}{n} \sum_{j=1}^{n} \sqrt{\text{LR}_j(\mathbf{X})} \leq \sqrt{\frac{1}{n} \sum_{j=1}^{n} \text{LR}_j(\mathbf{X})} = \text{LR}(\mathbf{X}) \tag{26}$$

$$\text{LR}(\mathbf{X}) = \frac{1}{n} \sum_{j=1}^{n} \text{LR}_j(\mathbf{X}) \leq \frac{1}{n} \sum_{j=1}^{n} \text{NLR}_j(\mathbf{X}) = \text{NLR}(\mathbf{X}) \tag{27}$$

Combining the inequality yields the desired result.

$\square$

**Corollary 3.3.** For $X_1, \dots X_n$ mutually independent variables with all $X_i$ centered unit variance random variables, it follows that $0 = \text{AAC}(\mathbf{X}) = \sqrt{\text{LR}(\mathbf{X})} = \sqrt{\text{NLR}(\mathbf{X})}$

*Proof of Corollary 3.3.* By Lemma 3.1 and Theorem 3.2 it suffices to show that if $X_1, \dots X_n$ are mutually independent variables with all $X_i$ centered unit variance that $\text{NLR}_j(\mathbf{X}) = 0$ for all $j$. This follows from

$$\text{NLR}_j(\mathbf{X}) = 1 - \min_{\theta} \mathbb{E}\left[ \|R_\theta(\mathbf{X}_{-j}) - X_j\|_2^2 \right] \tag{28}$$

$$= 1 - \min_{\theta} \left( \mathbb{E}\left[ R_\theta(\mathbf{X}_{-j})^2 \right] - 2\mathbb{E}\left[ R_\theta(\mathbf{X}_{-j})X_j \right] + \text{Var}[X_j] \right) \tag{29}$$

$$= - \min_{\theta} \left( \mathbb{E}\left[ R_\theta(\mathbf{X}_{-j})^2 \right] \right) = 0 \tag{30}$$

where we have used the assumed independence at $\mathbb{E}[R_\theta(\mathbf{X}_{-j})X_j] = \mathbb{E}[R_\theta(\mathbf{X}_{-j})]\,\mathbb{E}[X_j]$ and minimized the expression by using the predictor $R_0(\mathbf{X}) = 0$. As this holds for any $j$ we conclude that $\text{NLR}(\mathbf{X}) = 0$ from which the corollary follows. $\square$

**Corollary 3.4.** For $X_1, \dots, X_n$ centered random variables with unit variance, it holds that

$$\text{AAC}([X_1, \dots, X_n]) = 0 \iff \text{LR}([X_1, \dots, X_n]) = 0 \tag{6}$$

*Proof of Corollary 3.4.* ( $\Longleftarrow$ )

By Theorem 3.2 if $\text{LR}([X_1, \dots, X_n]) = 0$ then so must $\text{AAC}([X_1, \dots, X_n]) = 0$.

( $\Longrightarrow$ ) Given $\text{AAC}([X_1, \dots, X_n]) = 0$ all pairwise covariances $\text{Cov}(X_i, X_j)$ must be zero.

$$\text{LR}_j([X_1, \ldots, X_n]) = 1 - \min_{\vec{b}} \mathbb{E}\left[ \| \sum_{i=1, i \neq j}^{n} b_i X_i - X_j \|_2^2 \right] \tag{31}$$

$$= 1 - \min_{\vec{b}} \left( \mathbb{E}\left[ \sum_{i=1, i \neq j}^{n} \sum_{k=1, k \neq j}^{n} b_i b_k X_i X_k \right] - 2\mathbb{E}\left[ \sum_{i=1, i \neq j}^{n} b_i X_i X_j \right] + \text{Var}[X_j] \right) \tag{32}$$

$$= -\min_{\vec{b}} \left( \sum_{i=1, i \neq j}^{n} \sum_{k=1, k \neq j}^{n} b_i b_k \mathbb{E}[X_i X_k] - 2 \sum_{i=1, i \neq j}^{n} b_i \mathbb{E}[X_i X_j] \right) \tag{33}$$

$$= -\min_{\vec{b}} \left( \sum_{i=1, i \neq j}^{n} b_i^2 \mathbb{E}[X_i^2] \right) = -\min_{\vec{b}} \left( \sum_{i=1, i \neq j}^{n} b_i^2 \cdot 1 \right) = 0 \tag{34}$$

as $\mathbb{E}[X_i X_j] = 0$ due to $\text{Cov}(X_i, X_j) = 0$ and the variables being centered. To minimize, we choose $\vec{b} = \vec{0}$. As $j$ was arbitrary, this holds for any $\text{LR}_j([X_1, \ldots, X_n])$ and hence $\text{LR}([X_1, \ldots, X_n]) = 0$ as desired. $\qquad \square$

**Corollary 3.5.** For $X_1, \ldots, X_n$ centered variables with unit variance, $\text{LR}([X_1, \ldots, X_n]) = 0 \not\Rightarrow \text{NLR}([X_1, \ldots, X_n]) = 0$

*Proof of Corollary 3.5.* We show a counterexample which proves the implication cannot hold. Let $X, Y \sim \mathcal{N}(0, 1)$. Further, let $Z = \frac{1}{2}\left(X^2 - Y^2\right)$. We have

$$\mathbb{E}[Z] = \mathbb{E}\left[ \frac{1}{2}\left(X^2 - Y^2\right) \right] = \frac{1}{2}\left( \mathbb{E}[X^2] - \mathbb{E}[Y^2] \right) = 0 \tag{35}$$

$$\text{Var}[Z] = \text{Var}\left[ \frac{1}{2}\left(X^2 - Y^2\right) \right] = \frac{1}{4}\left( \text{Var}[X^2] + \text{Var}[Y^2] \right) = \frac{1}{4}(3 - 1 + 3 - 1) = 1 \tag{36}$$

where we used the fact that the 4-th moment of a centered unit variance Gaussian is 3 and the second moment is 1.

For $\text{NLR}([X, Y, Z])$ we find

$$\text{NLR}([X, Y, Z]) = \frac{3 - \min_\theta \mathbb{E}\left[ \|R_\theta(X, Y) - Z\|_2^2 \right] - \min_\theta \mathbb{E}\left[ \|R_\theta(X, Z) - Y\|_2^2 \right] - \min_\theta \mathbb{E}\left[ \|R_\theta(Z, Y) - X\|_2^2 \right]}{3} = 1 \tag{37}$$

as given any two variables, a nonlinear predictor is able to perfectly reconstruct the third variable by construction of our example.

Given by assumption $\text{LR}([X_1, \ldots, X_n]) = 0$ we must by Corollary 3.4 also have $\text{AAC}([X_1, \ldots, X_n]) = 0$ from which we infer

$$\text{AAC}([X, Y, Z]) = \frac{|\text{Cov}(X, Y)| + |\text{Cov}(X, Z)| + |\text{Cov}(Y, Z)|}{3} = \frac{|\text{Cov}\left(X, \frac{1}{2}\left(X^2 - Y^2\right)\right)| + |\text{Cov}\left(Y, \frac{1}{2}\left(X^2 - Y^2\right)\right)|}{3} \tag{38}$$

$$= \frac{|\text{Cov}\left(X, X^2 - Y^2\right)| + |\text{Cov}\left(Y, X^2 - Y^2\right)|}{12} \tag{39}$$

$$= \frac{|\text{Cov}\left(X, X^2\right)| + |\text{Cov}\left(X, Y^2\right)| + |\text{Cov}\left(Y, X^2\right)| + |\text{Cov}\left(Y, Y^2\right)|}{12} = 0 \tag{40}$$

where we have used $\text{Cov}(X, Y) = 0$ as well as $\text{Cov}(X, Y^2) = 0$ which follow from the independence of the centered Gaussians $X$ and $Y$. Further $\text{Cov}(X^2, X) = \mathbb{E}[X^3] - \mathbb{E}[X]\mathbb{E}[X^2] = 0$.

Which shows that although LR and NLR are zero NLR takes its maximal value for this example. This proves the Corollary by counterexample. $\qquad \square$

# E    Further Implementation Details

Certain parameters such as masking fraction, $\lambda$ and the ridge penalty were determined by a grid search over a logarithmic search space. Most of these results are included in the ablations.

For optimization, the LARS optimizer is used (You et al., 2017) as well as a linear warm-up cosine annealing learning rate scheduler. Learning rates can be found in the accompanying code.

For all models except SSLPM-SGD a batch size of 256 images was used. In SSLPM-SGD we have used 512 to increase numerical stability.

SSLPM-SGD is trained with the AdamW (Loshchilov & Hutter, 2019) optimizer using a base learning rate of $1e - 3$ a weight decay of $1e - 6$.

Further, different methods to enhance the predictor training stability were used:

- The predictor does not have minibatches, we optimize with all data per batch.

- Instead of having the encoder optimize $\mathcal{L}_{\text{invariance}} - \lambda \mathcal{L}_{\text{pred}}$ we optimize $\mathcal{L}_{\text{invariance}} - \lambda \log\left(\mathcal{L}_{\text{pred}}\right)$.

- For every optimization step of the encoder, the predictor can perform up to 500 optimization steps, but stopping early if MSE stops improving on the validation mask.

- We vary the learning rate of the predictor. For the first 25 batches, we increase the learning rate by multiplying with 1.005. Afterward, we decrease it by dividing by 1.005.

- The predictability loss is clipped at 1 from above.

A single training run of the models presented takes between 8 and 24 hours (depending on the model and hyperparameters) on a modern GPU such as a NVIDIA RTX3090. Evaluating the redundancy takes around 1 hour per trained model.

The ImageNet-100 dataset is the same as used in the solo-learn paper (Da Costa et al., 2022) and is a 100 category subset of the ImageNet dataset Deng et al. (2009).

# F    Additional Evidence for Analysis & Results

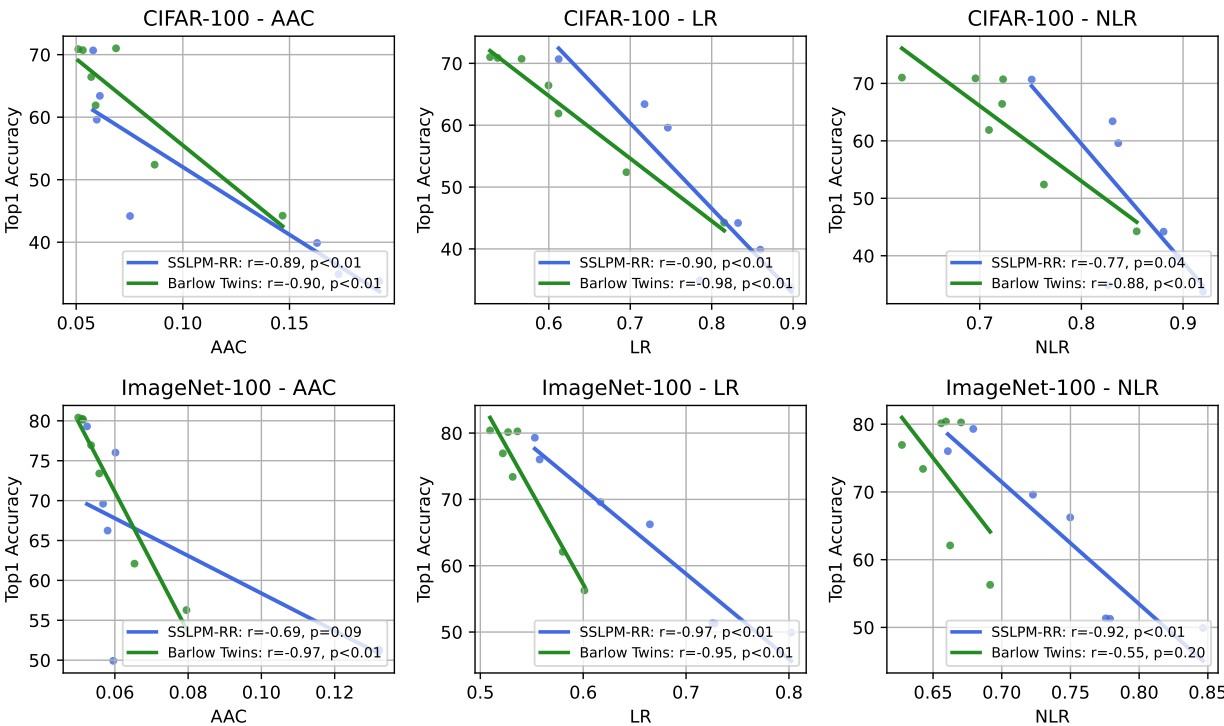

Figure 13: Relationship between Top-1 accuracy and different redundancy measures redundancies on CIFAR-100 and ImageNet-100 plotted with the best fit line. For the best fit line Pearson correlation and $p$-value for the null hypothesis that the data is uncorrelated is reported.

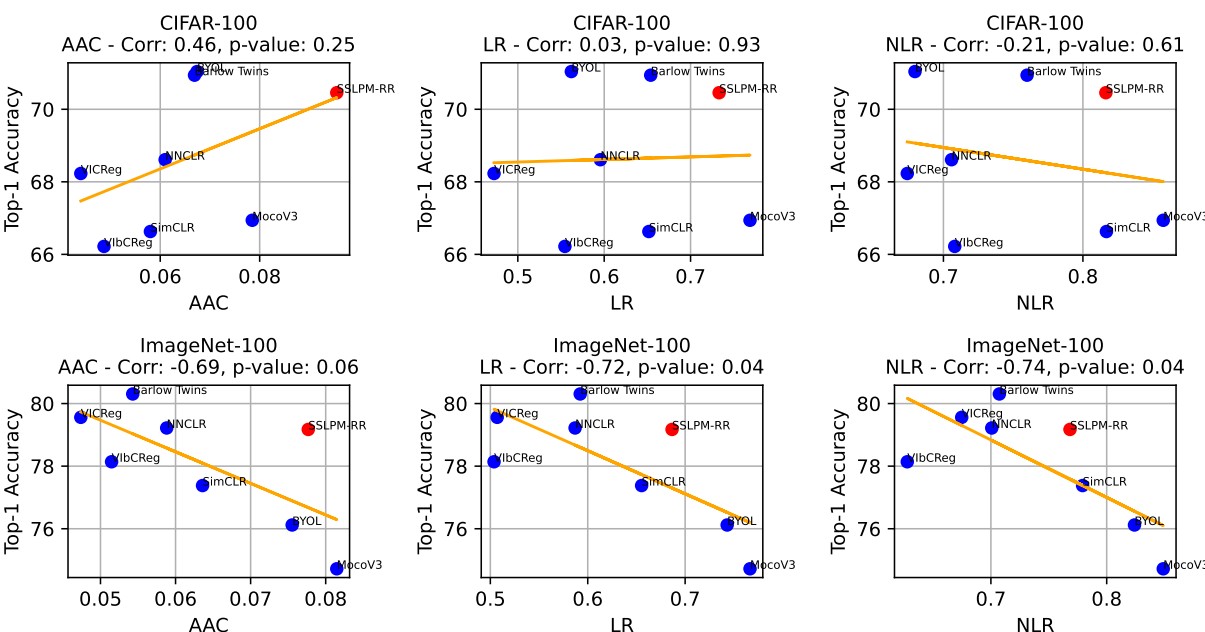

Figure 14: Redundancy measures vs. Top-1 accuracy of SSL methods on CIFAR-100, and ImageNet-100. For the best fit line Pearson correlation and $p$-value for the null hypothesis that the data is uncorrelated is reported.

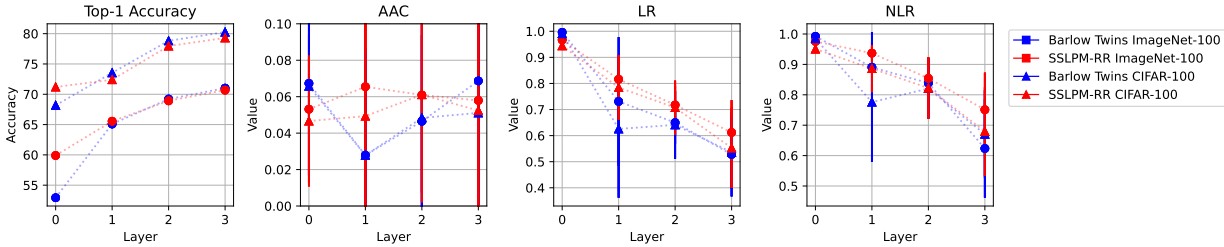

Figure 15: Ablation on the number of layers of the projector for CIFAR-100 and ImageNet-100. Redundancy measures have the standard deviation over all measured indices shown.

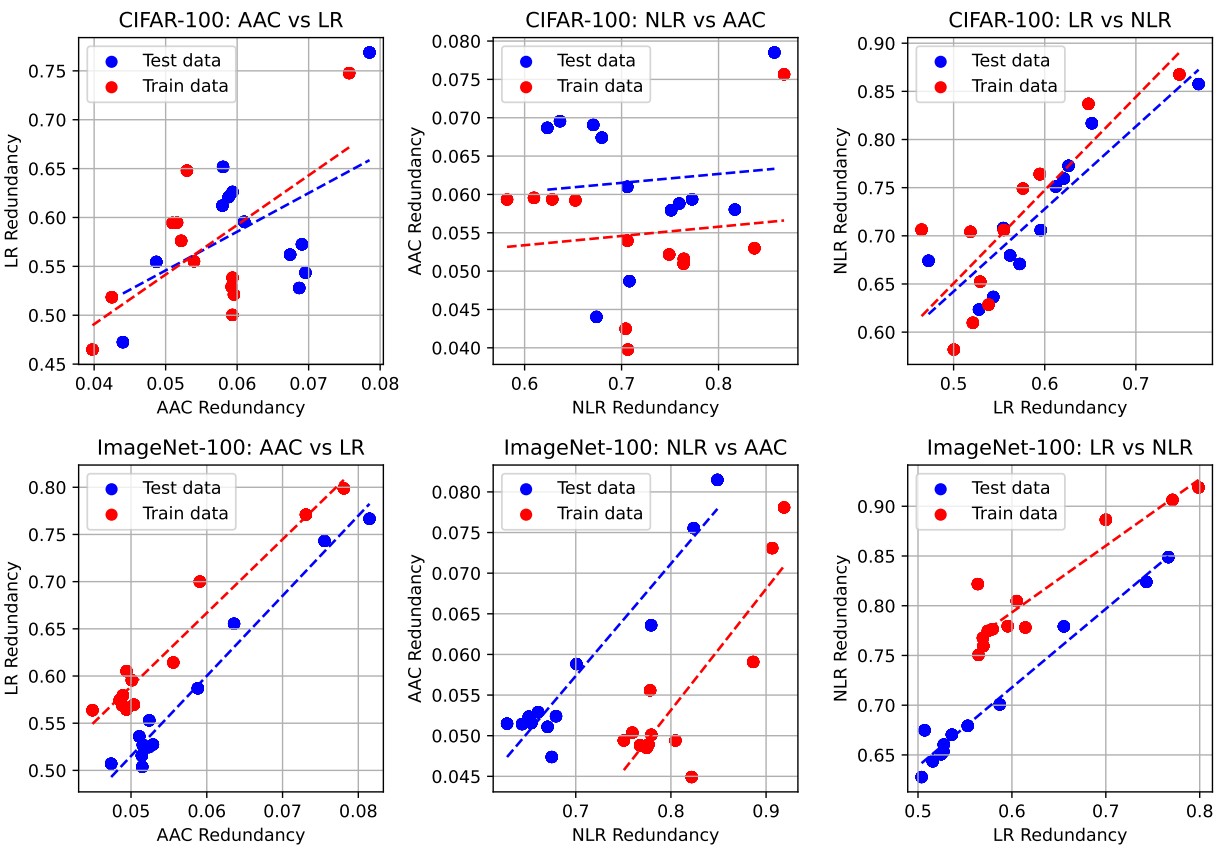

Figure 16: Difference in redundancy between test and train sets for CIFAR-100 and ImageNet-100 embeddings.

