# OpenReview forum: "Beyond Pairwise Correlations: Higher-Order Redundancies in Self-Supervised Representation Learning"
_TMLR — Rejected by TMLR_

### Review · Reviewer_3poe · 2025-02-21

**Summary Of Contributions:**

This paper explores higher-order redundancies loss functions beyond second order in self-supervised, with Barlow Twins as a baseline. Several redundancy reduction measures are presented, from which several loss functions are derived, leading to new Self-Supervised Learning SSL methods. The presented methods are evaluated on linear evaluation on CIFAR and ImageNet-100 and compared to other SSL baselines including Barlow Twins. Additional ablations study the correlation between redundancy reduction and performance.

**Audience:**

Yes

**Claims And Evidence:**

No

**Requested Changes:**

Provide experiments highlighting the advantage of higher order methods in SSL compared to second order redundancy reduction methods.

**Strengths And Weaknesses:**

Strengths:
- The idea of studying higher order redundancy reduction methods is interesting, and is a valid direction to explore for learning better self-supervised representations.

- The paper introduces several interesting redundancy measures that could be useful as a practical tool for representation learning.

Weaknesses:
- Unfortunately, the initial claims made by the paper are not supported by the experiments, and the findings are negative. The main experiment consists in evaluating the quality of the learned features on linear evaluation on CIFAR and ImageNet-100. These datasets  are very small and toy, especially compared to most works from the last years of SSL literature which focused on ImageNet, and even of these, there is no performance improvement over Barlow Twins. This shows that higher order loss functions do not lead to learning better representations.

- The other experiments in the paper study the correlations between redundancy and performance, but are not very convincing and do not point to any meaningful conclusion on whether removing higher order redundancy helps capturing more information. Other tasks than linear evaluation could have been experimented, such as out of domain generalization.

- Many variations of the same method are presented, SSLPM-RR, SSLPM-SGD, SSLPM, but it’s hard to understand behind having all these variations. I would suggest to focus on one method and focus on the comparison between second-order (Barlow Twins) and higher order (You method), and finding a practical setting where yours has a competitive advantage.

---

> ### Author Response · Authors · 2025-04-08
> **Response to Reviewer 3poe**
>
> Thank you for your time and detailed feedback.
>
> We would like to address the concerns raised, particularly regarding the findings and the experimental setup.
>
> **1. Goal of the Study: Understanding over SOTA Performance**
>
> You noted that our proposed methods do not show a performance improvement over Barlow Twins  and concluded that higher-order loss functions do not lead to better representations. We agree with this empirical observation; indeed, one of our main findings is that reducing more complex, higher-order redundancies did not yield improved downstream performance in our experiments.
>
> However, we would like to emphasize that the primary objective of this work was not to achieve new state-of-the-art performance, but to deepen the understanding of redundancy in SSL embedding spaces. We aimed to formalize notions of higher-order redundancy, analyze their properties, measure them empirically across various SSL methods, and investigate the effect of explicitly minimizing them. Discovering that minimizing these specific higher-order redundancies doesn't boost performance is a key part of the gained understanding.
>
> **2. Dataset Choice and Experimental Rigor**
>
> You raised concerns about the use of CIFAR-10/100 and ImageNet-100, considering them "small and toy" compared to larger datasets often used in SSL literature. We chose these datasets deliberately. Because our focus was on understanding the nuanced effects of different redundancy reduction strategies, ensuring a fair and rigorous comparison required extensive hyperparameter tuning. Drawing insights from comparisons involving suboptimally tuned models would likely not be valuable, and compute restrictions on our end made this tuning infeasible on large datasets.
>
> **3. Correlation Experiments and Method Variations**
>
> The correlation experiments were central to our goal of understanding redundancy's role. We found that for Barlow Twins and SSLPM-RR, which explicitly target redundancy, lower linear redundancy (LR) consistently correlated with higher performance. Yet, looking across diverse SSL methods revealed only a weak or inconsistent correlation overall. This contrast itself is a key finding: the impact of measurable redundancy on performance seems highly dependent on the specific SSL method's mechanics, rather than being a simple universal principle. This complexity underscores why a deeper investigation, rather than solely pursuing SOTA, was warranted.
>
> SSLPM-RR (using ridge regression for linear predictability) and SSLPM-SGD (using an MLP for non-linear predictability) were necessary variations to probe the impact of minimizing different kinds of redundancy. The results showing SSLPM-RR performing better and SSLPM-SGD performing worse with more complex predictors directly support our conclusion about the limited benefit of minimizing the higher-order redundancies we investigated."
>
>
> **4. Addressing the Requested Change**
>
> You requested experiments highlighting the advantage of higher-order methods. As discussed above, our experiments consistently showed no such performance advantage. The contribution lies not in demonstrating an advantage, but in the systematic investigation itself, the formalization of the concepts, and the finding that simply reducing more complex forms of redundancy does not automatically improve representation quality, which we believe is an important insight for the field.
>
> In conclusion, our work aimed to explore the uncharted territory beyond pairwise correlations in SSL redundancy reduction. While we did not find that minimizing the studied higher-order redundancies leads to performance gains, we provided a framework for analyzing these redundancies and offered empirical insights into their complex relationship with performance across various methods.
>
> Thank you again for your valuable feedback. Please let us know if you have any questions!

---

### Review · Reviewer_sgcx · 2025-02-27

**Summary Of Contributions:**

The paper explores the idea of penalizing different redundancy measures in SSL methods like Barlow Twins. The authors propose a couple redundancy measures with increasing generality, including pair-wise prediction via correlation, linear prediction via ridge regression, and nonlinear prediction with a masked MLP. The experiments show that minimizing these redundancy measures alongside the standard augmentation invariance loss does not improve performance.

**Audience:**

Yes

**Broader Impact Concerns:**

None.

**Claims And Evidence:**

Yes

**Requested Changes:**

Several questions and requested changes are mentioned above.

**Strengths And Weaknesses:**

## Strengths

The paper makes a reasonable hypothesis and tests it. A couple versions of the proposal are tested, and the implementation details are reasonably described. Although the results do not support the proposal, it may be useful to publish this failed direction in case others want to consider it.

## Weaknesses

- Barlow Twins is not a SOTA SSL method these days, so it's not the highest priority to build upon. But it's probably okay and the paper should still be of interest to some readers.

- In addition to not helping, all the proposed methods add complexity and computational cost on top of the original Barlow Twins formulation. They therefore seem worse in multiple dimensions.

- The experiments are small-scale, both in terms of datasets and models. But that may be okay in this case, since there aren't obvious concerns of results changing with scale.

- There's a fairly simple argument to be made that any successful pre-training method should have decorrelated embedding dimensions: representations are bottlenecked by the embedding size, and having high correlation/predictability means the network wastes dimensions. It might be good to discuss this point, and perhaps test redundancy measures and other networks (e.g., CLIP, SimCLR).

- In eq. 1, it's a bit confusing to describe $X_A^\top X_B$ as a "correlation matrix." I get the vague intuition, but it's not clear what the random variable is here. In subsequent discussions, we let each dimension be a random variable and this makes more sense.

- It might be worth mentioning that when we calculate $Cov(X_i, X_j)$ in $AAC_j(X)$, we're doing something similar to $LR_j(X)$: the squared correlation is the MSE of a univariate linear predictor.

- For SSLPM-SGD, it would be good to comment on the relative cost of optimizing the reconstruction network over multiple steps before taking each step on the encoder. The description of the architecture is also inconsistent: Section 3.4.1 says the layer widths are 512-128-64-1, but the output should be the same size as the input, and the experiments later say we use either 1 or 3 hidden layers, not 2.

- For SSLPM-RR, it would help to explain whether you backpropagate through the optimal $W$. Could this choice make a difference?

- In Figure 3, it's concerning how sensitive performance is to the choice of $\lambda$. It might be interesting to know if this is the same for the original Barlow Twins formulation.

- Am I missing something, or was $AAC(X)$ not used as a redundancy measure in any training runs? I see it quantified for a set of models in Figures 4-5, but I can't tell if it was used during training. Would this be worth adding as an additional method?

---

> ### Author Response · Authors · 2025-04-07
> **Response to Reviewer sgcx**
>
> We thank the reviewer for their very valuable and constructive feedback
>
> Below, we address the issues raised:
>
> **Barlow Twins is not a SOTA SSL method these days, so it's not the highest priority to build upon. But it's probably okay and the paper should still be of interest to some readers.**
>
> We acknowledge that Barlow Twins is not the current SOTA method, but we chose it as our baseline because it provides a clear framework for studying redundancy reduction through its explicit mechanism. Our work focuses on investigating the role of redundancy in SSL rather than pushing the performance frontier.
>
> **In addition to not helping, all the proposed methods add complexity and computational cost on top of the original Barlow Twins formulation.**
>
> It is true that there is additional complexity introduced:
>
> - In the case of SSLPM-RR training takes around 3% longer than the standard Barlow Twins.
> - For the SSLPM-SGD things are not as clear-cut - we used a different graphics card and due to a bug on our infrastructure did not enable half precision training. But from our internal profiling, we know that roughly 10% of the time spent is used to optimize the predictor.
>
> However, we are not aiming for a new state of the art but interested in shedding some light on the role of redundancy reduction in SSL.
>
> **The experiments are small-scale, both in terms of datasets and models. But that may be okay in this case, since there aren't obvious concerns of results changing with scale.**
>
> We appreciate your understanding on this point - unfortunately, we are compute limited on our end given the hyperparameter tuning involved to make all models comparable on large datasets.
>
> **There's a fairly simple argument to be made that any successful pre-training method should have decorrelated embedding dimensions: representations are bottlenecked by the embedding size, and having high correlation/predictability means the network wastes dimensions. It might be good to discuss this point, and perhaps test redundancy measures and other networks (e.g., CLIP, SimCLR).**
>
> See Figure 7 of the paper and Figure 14 in the appendix. As you suggested, we examined redundancy across different networks and found that all methods outperforming our SSLPM-RR exhibit less redundancy in their embeddings, supporting your argument about successful pre-training methods having decorrelated embedding spaces.
>
> **In eq. 1, it's a bit confusing to describe X^T X as a "correlation matrix." I get the vague intuition, but it's not clear what the random variable is here.**
>
> Thanks for this input, we have renamed it to "dot product matrix."
>
> **It might be worth mentioning that when we calculate Cov(Xi, Xj) in AACj(X), we're doing something similar to LRj(X): the squared correlation is the MSE of a univariate linear predictor.**
>
> We do mention this and we also give an example (Example 3.1 on page 5) on how they are different.
> Moreover, LR correlates well with performance for Barlow and SSLPM (whereas AAC does not) so LR might be a better/more useful measure of redundancy than AAC.
>
> **For SSLPM-SGD, it would be good to comment on the relative cost of optimizing the reconstruction network over multiple steps before taking each step on the encoder. The description of the architecture is also inconsistent...**
>
> This is an omission from our end - thank you for catching this. We have fixed this in the paper.
>
> **For SSLPM-RR, it would help to explain whether you backpropagate through the optimal W. Could this choice make a difference?**
>
> As W can be solved using a linear system of equations we use least squares to solve directly - and then when training the encoder with this additional loss, we do backpropagate through W since the loss is calculated using W. The weights W are frozen during the update step. See equation (12). We hope this clarifies this question, but please let us know if we haven't explained it clearly.
>
> **In Figure 3, it's concerning how sensitive performance is to the choice of λ. It might be interesting to know if this is the same for the original Barlow Twins formulation.**
>
> In our implementation of CLPM-RR we define `total_loss = diag_loss - lamb * (proj_output_dim * prediction_loss)` whereas for Barlow Twins this multiplication is not performed. Hence, we believe they cannot be compared 1:1. We did similar investigations for Barlow Twins but see model collapse when lambda is too large instead of slowly decreasing performance.
>
> **Am I missing something, or was $\operatorname{AAC}$ not used as a redundancy measure in any training runs**
>
> We did indeed not run any runs, as it is highly similar to the invariance loss from the Barlow Twins method (absolutes instead of square). Upon your request, we did run experiments on CIFAR-10 yielding a top-1 accuracy of 87.39 - so a bit below of what we see with the other methods - though this could also be due to us just optimizing the lambda and keeping all other parameters fixed from the Barlow Twins case.

---

### Review · Reviewer_5xLo · 2025-03-10

**Summary Of Contributions:**

This paper studied self-supervised learning from redundancy reduction perspective. The authors first defined the redundancy in SSL embedding space with average absolute covariance and predictability. An SSL method  Self Supervised Learning with Predictability Minimization (SSLPM) was proposed to reduce redundancy in the embedding space via minimizing redundancy reduction loss and predictability of one dimension of embedding with the rest dimensions. The experiments showed that SSLPM is competitive with state-of-the-art methods and the current SOTA SSL methods can implicitly reduce the redundancy.

**Audience:**

Yes

**Broader Impact Concerns:**

No broader impact concerns.

**Claims And Evidence:**

Yes

**Requested Changes:**

While the study explores various aspects of redundancy reduction in SSL, it doesn't provide clear, actionable guidelines for practitioners on how to optimally implement these insights in different SSL scenarios. SSLPM seems not a correct solution. The authors should consider how to prompt the paper from this perspective.

**Strengths And Weaknesses:**

Strenghts:

- The paper addresses a crucial issue in Self-Supervised Learning (SSL): the redundancy of the pretrained embedding space. This topic is significant for improving the efficiency and effectiveness of SSL models.

- The paper offers a comprehensive analysis of the role of redundancy reduction in SSL, exploring its effects under various conditions and model configurations. These analysis are supported by extensive experimentation, providing a solid empirical foundation for its arguments and findings.

Weakness:

- The paper's main contribution lacks clarity and coherence. While it presents several observations and findings, these don't coalesce into a unified narrative. The importance of redundancy reduction appears inconsistent across different scenarios, without a clear statement of when it should be considered crucial.

- The main findings of the paper, which is summarized in the end of the introduction are not novel. Similar findings regarding strength of redundancy reduction ($\lambda$) and the depth of projection layers can be found also in Barlow Twin paper.

- The author proposed SSLPM method, while it does not demonstrate a better performance or presents extra findings other than Barlow Twins. It is unclear why SSLPM is necessary for the paper.

- The SSLPM-RR variant appears redundant given that Barlow Twins already minimizes correlation in the embedding space. It's unclear how SSLPM-RR provides additional benefits beyond what Barlow Twins achieves in terms of promoting independence between embedding dimensions.

---

> ### Author Response · Authors · 2025-04-08
> **Response to Reviewer 5xLo**
>
> Thank you for your detailed review and constructive feedback.
>
> We have revised parts of the framing of the paper aiming to address your concerns regarding clarity, novelty, the role of SSLPM, and to some degree actionable guidelines.
>
> **Novelty of Findings:**
>
> While lambda tuning and projector effects are known concepts, our work provides novel analysis specifically concerning their impact on higher-order redundancy measures (LR, NLR) within our predictability minimization framework
>
> Several core findings are novel contributions.
> - The direct empirical evidence that minimizing the non-linear redundancy (NLR) we defined does not boost performance.
> - Actively minimizing NLR – a non-trivial extension of existing ideas – does not yield benefits is itself a valuable contribution.
> - Link between linear redundancy (LR) and performance for Barlow Twins and SSLPM-RR,
> - The observation of implicit redundancy reduction versus weak general correlation across diverse SSL methods.
>
> **Necessity and Role of SSLPM / SSLPM-RR:**
>
> Regarding SSLPM-RR's distinction from Barlow Twins: while Barlow Twins targets pairwise correlations, SSLPM-RR targets linear predictability from all other features. These are mathematically distinct, as shown in Example 3.1, allowing us to specifically investigate the impact of LR reduction. The strong correlation found between LR and performance (Fig. 4 ) further validates investigating this specific measure.
>
> **Actionable Guidelines & Reframing:**
> Our work does indeed not yield simple, universal rules, reflecting the complexity found. However, we believe the understanding gained is fundamentally interesting: Insights like the importance of LR for Barlow-like methods, the diminishing returns from minimizing NLR, and the role of projector depth on LR/NLR are of interest for follow-up research as novel ideas are needed as reducing nonlinear redundancies is (a) non-trivial, as well as (b) likely not fruitious. We have focused on "prompting the paper from this perspective" through the subtle narrative reframing in the revised Abstract, Introduction, Methods, and Conclusion.
>
> Thank you again for your thorough review. Please let us know if anything is unclear on not well addressed.

---

### Decision · Action_Editor_GFJw · 2025-04-30

**Recommendation:** Reject

**Comment:**

According to the reviewers, several issues remain unresolved in the updated version. In particular, two reviewers expressed disappointment that their comments and suggestions were not fully addressed. Additionally, multiple reviewers noted in their final recommendations that the overall narrative of the paper lacks clarity and that "some general rules and findings are not explained."

The Area Chair finds these concerns valid and recommends rejection of the current version. The authors are encouraged to address the reviewers' feedback more thoroughly and consider resubmitting the paper with major revisions.

**Audience:**

This is an active area with a large audience.

**Claims And Evidence:**

According to 3 qualified reviewers the current manuscript of this submission does not contain sufficient evidence to support the paper's claims. They also believe that there is clear room for improvement

**Resubmission Of Major Revision:**

The authors may consider submitting a major revision at a later time.